# VaiPhy: a Variational Inference Based Algorithm for Phylogeny

**Hazal Koptagel**[12*]    **Oskar Kviman**[12*]    **Harald Melin**[12]
**Negar Safinianaini**[12]    **Jens Lagergren**[12]

[1]School of EECS, KTH Royal Institute of Technology, Stockholm, Sweden
[2]Science for Life Laboratory, Solna, Sweden
*Equal contribution
{koptagel,okviman,haraldme,negars,jensl}@kth.se

## Abstract

Phylogenetics is a classical methodology in computational biology that today has become highly relevant for medical investigation of single-cell data, e.g., in the context of cancer development. The exponential size of the tree space is, unfortunately, a substantial obstacle for Bayesian phylogenetic inference using Markov chain Monte Carlo based methods since these rely on local operations. And although more recent variational inference (VI) based methods offer speed improvements, they rely on expensive auto-differentiation operations for learning the variational parameters. We propose VaiPhy, a remarkably fast VI based algorithm for approximate posterior inference in an *augmented tree space*. VaiPhy produces marginal log-likelihood estimates on par with the state-of-the-art methods on real data and is considerably faster since it does not require auto-differentiation. Instead, VaiPhy combines coordinate ascent update equations with two novel sampling schemes: (i) *SLANTIS*, a proposal distribution for tree topologies in the augmented tree space, and (ii) the *JC sampler*, to the best of our knowledge, the first-ever scheme for sampling branch lengths directly from the popular Jukes-Cantor model. We compare VaiPhy in terms of density estimation and runtime. Additionally, we evaluate the reproducibility of the baselines. We provide our code on GitHub: https://github.com/Lagergren-Lab/VaiPhy.

## 1 Introduction

Phylogenetic software has for a long time been applied in biological research, and biological findings relying on phylogenetic trees are frequent. Moreover, due to the emergence of single-cell sequencing, phylogenetic inference for bifurcating and multifurcating trees can now be utilized in medical studies of how cells differentiate during development and how tumor cells progress in cancer.

When attempting to infer the posterior distribution over phylogenetic trees for $|X|$ taxa, represented as sequences, the main obstacle is the exponential size of the tree space, i.e., the set of candidate trees having those taxa as leaves. In Markov chain Monte Carlo (MCMC) based Bayesian phylogenetic inference, e.g., MrBayes [28] and RevBayes [20], this tree space is explored by performing local operations on the trees. In order to avoid local operations, Bouchard-Côté et al. [3] launched Sequential Monte Carlo (SMC) as an alternative methodology for Bayesian phylogenetic inference.

Probabilistic phylogenetic inference is a machine learning problem, and the key to unlocking this crucial application can be expected to be found in the machine learning methodology; in particular, considering the capacity of the VI methodology, [32, 1], to deliver impressive performance gains for Bayesian inference, e.g., compared to MCMC. Expectedly, such gains have recently been made within the realm of phylogenetic analysis.

36th Conference on Neural Information Processing Systems (NeurIPS 2022).

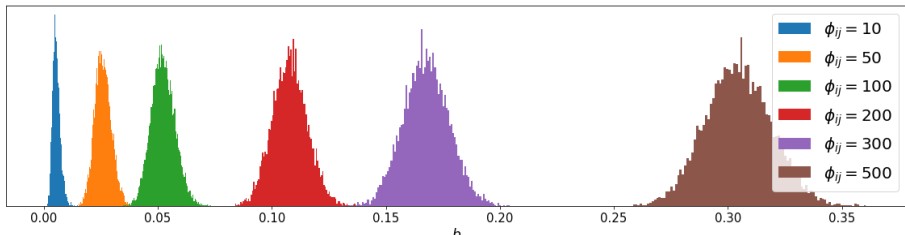

Figure 1: Visualization of branch lengths, $b$, sampled directly from the Jukes-Cantor model. The figure shows samples for different numbers of estimated mutations along an edge, $\phi_{i,j}$, when the total number of sites are $M = 2000$.

For instance, inference based on the computationally costly CAT model, which allows different equilibrium frequencies across sequence sites, has been made substantially more efficient by the application of VI, [5]. Also, [11] applied VI to render the analysis of a sequence evolution model more efficient. They consider the efficiency improvements obtained by VI, in particular using the generic STAN software [11], when analyzing a single fixed tree topology. Interestingly, the VI approach can also be used to analyze a large number of candidate phylogenetic trees. In [38], so-called subsplit Bayesian networks (SBNs) were introduced in order to represent the posterior over a given set of candidate trees. This approach has subsequently been developed further by introducing an edge (a.k.a. branch) length distribution [39] and making the edge length distribution more expressive by taking advantage of normalizing flows [37]. In particular, [39] obtains posterior distributions that improve on the marginal likelihood obtained by using MrBayes, [21], in combination with the improved Stepping Stone (SS) method, [8].

The SBN approach evades the exponentially sized tree space by considering a given set of candidate trees and the partitions of the given taxa defined by those candidate trees. It is noteworthy that the VI methodology earlier has been successfully applied to bipartite matching and a few other combinatorial problems [2]; the phylogeny problem, however, involves both combinatorial and continuous parameters.

Multifurcating trees are common in biology, and biologists are well familiar with them. For instance, in a clone tree that represents the evolutionary relationships between the subclones of a tumor, nodes may have more than two children. Even in cases with no biological support for multifurcating trees, biologists are used to working with multifurcating trees. E.g., the consensus method used to analyze the posterior of a MrBayes run may produce a multifurcating tree.

**Contributions:** We present a novel framework for phylogenetic inference that produces marginal likelihood estimates on par with the state-of-the-art VI methods while being up to 25 times faster than VBPI-NF on real datasets. The framework is on the top level a mean-field VI algorithm and is comprised of the following three novel methods:

- **SLANTIS** is a proposal distribution that proposes multifurcating spanning trees. Sampling from SLANTIS is performed following an approach suggested by [6] for sampling a perfect matching in bipartite graphs: it is fast, and the sampled tree's compensation weight is easily computed without additional cost of time.

- **The JC sampler** is a novel scheme for sampling branch lengths directly from the Jukes-Cantor (JC69) model [22]. This is a significant contribution, as no previous sampling scheme is known for this task and the JC model arguably is a commonly used substitution model in phylogenetics.

- **VaiPhy** is a VI algorithm for obtaining posterior approximations over the set of branch lengths, the ancestral sequences, and tree topologies, given the leaf set observations. VaiPhy utilizes SLANTIS and the JC sampler in order to compute the importance weighted evidence lower bound (IWELBO; [4]) and its update equations. Compared to existing VI methods in phylogenetics, VaiPhy extends the target distribution by considering an auxiliary variable, the ancestral sequences, and it does not require any input consisting of precomputed trees.

We demonstrate the flexibility of our framework by using its learned parameters to parameterize two new proposal distributions for the Combinatorial SMC (CSMC) algorithm. We compare our CSMC

method, $\phi$-CSMC, with the other two main SMC based methods, outperforming them in terms of marginal likelihood estimation on real data.

Finally, we contribute to the field of Bayesian phylogenetic inference by benchmarking the most prominent methods in the field on seven popular real datasets.

## 2  Background

In the Bayesian phylogenetic setting, we are interested in the posterior distribution over phylogenies – pairs of tree topologies and sets of branch lengths – given observations on the leaves,

$$p_\theta(\tau, \mathcal{B}|X) = \frac{p_\theta(\tau, \mathcal{B}, X)}{p_\theta(X)}, \tag{1}$$

where $\tau$, $\mathcal{B}$, and $X$ denote the tree topology, the set of branch lengths and the observations, respectively. More specifically, $\mathcal{B}$ contains a single branch length per edge, $e$, in the set of edges in $\tau$, i.e., $\mathcal{B} = \{b(e) : e \in E(\tau)\}$. Unfortunately, $p_\theta(\tau, \mathcal{B}|X)$ is intractable, so we need to approximate it using Bayesian inference. Before going into how to approximate this quantity, we first provide some crucial notation.

In phylogenetics, one is often concerned with bifurcating $X$-trees. These trees have labeled leaf nodes, and their internal vertices have either two (root in a rooted tree) or three degrees. Meanwhile, multifurcating trees' internal vertices can have higher degrees. We use $M$, $\Sigma$, and $N$ to denote the sequence length, the set of nucleotides, and the number of vertices, respectively. Further, we let $|X|$ be the number of given taxa, $V$ and $E$ be the set of vertices and edges, while $(V, E)$ is an acyclic graph.

Let $Z_i^m$ be the probabilistic ancestral sequence of node $i$ at position $m$, and $\theta$ denote the model parameters, such as the branch length prior parameters or the parameter choices in the substitution model. Then, the evolutionary model along an edge, $e = (i, j)$, is $p_\theta(Z_i^m = \alpha | Z_j^m = \beta) = exp(b(e)\ Q_{\alpha\beta})$, where $Q$ is the rate matrix, parameterized according to the JC69 model, and $\alpha, \beta \in \Sigma$ are nucleotides. The likelihood is defined as

$$p_\theta(X|\tau, \mathcal{B}) = \sum_{Z \in \Sigma^M} p_\theta(X, Z|\tau, \mathcal{B}),$$

and can be computed effectively using Felsenstein's pruning algorithm [9].

## 3  Proposed Framework

Existing Bayesian phylogenetic approaches often perform inference in a sample space over bifurcating $X$-trees and branch length sets, $\Omega$. In contrast, we propose to work in an augmented sample space, A. Firstly, we work in a tree topology space for multifurcating trees with labeled internal vertices. Secondly, we extend the target distribution in Eq. (1) by considering ancestral sequences on the internal vertices as an auxiliary variable, $Z$.

Apart from that, our augmented space offers higher modeling flexibility (clearly, since $\Omega \subset$ A); it enables posterior estimation over $Z$, which further lets us estimate the number of mutations between all vertices in the sample space. This is a key insight that ultimately allows us to devise VaiPhy, a mean-field coordinate ascent VI [19] algorithm.

### 3.1  VaiPhy

In the augmented space, A, we wish to approximate the posterior distribution

$$p_\theta(\tau, \mathcal{B}, Z|X) \propto p_\theta(X, Z|\mathcal{B}, \tau)p_\theta(\mathcal{B}|\tau)p_\theta(\tau), \tag{2}$$

using a factorized variational distribution

$$q(\tau, \mathcal{B}, Z|X) = q(\mathcal{B}|X)q(\tau|X)q(Z|X), \tag{3}$$

where $q(Z|X) = \prod_{i \in I(A)} q(Z_i|X)$. Here, $I(A)$ denotes the set of internal vertices in the augmented space. We will use $q(Z|X)$ to compute $\phi_{ij} \in [0, M]$, the expected number of mutations over the edge $(i, j)$, for all possible edges. Notice that the maximum number of observed mutations between two vertices is the number of sites, $M$.

As is standard in classical VI, we alternatively update the distributions in Eq. (3) in order to maximize the evidence lower bound (ELBO),

$$\mathcal{L} = \mathbb{E}_{q(\tau, \mathcal{B}, Z|X)} \left[ \log \frac{p_\theta(X, Z|\mathcal{B}, \tau)p_\theta(\mathcal{B}|\tau)p_\theta(\tau)}{q(\mathcal{B}|X)q(\tau|X)q(Z|X)} \right], \tag{4}$$

over iterations. Each iteration, we also update $\phi$, following [12]

$$\phi_{ij} = \sum_{\alpha \neq \beta} F_{ij}^{\alpha, \beta}, \tag{5}$$

where $F_{ij}^{\alpha, \beta}$ is the estimated number of times a nucleotide mutates from $\alpha$ to $\beta$, defined as $F_{ij}^{\alpha, \beta} = \sum_{m=1}^{M} q(Z_i^m = \alpha|X^m)q(Z_j^m = \beta|X^m)$.

The update equations for the variational distributions are derived in a mean-field fashion. Starting with $q(Z_i|X)$, we find that

$$\log q^*(Z_i|X) \stackrel{\pm}{=} \mathbb{E}_{q(\tau, \mathcal{B}, Z_{\neg i}|X)} [\log p_\theta(X, Z, \mathcal{B}, \tau)]$$
$$\stackrel{\pm}{=} \mathbb{E}_{q(\tau, \mathcal{B}|X)} \left[ \sum_{Y_j \in Y_{N_\tau(i)}} q(Y_j|X) \log p_\theta(Y_j|Z_i, b(i, j)) \right], \tag{6}$$

where $\stackrel{\pm}{=}$ denotes equal up to an additive constant, $Y_{N_\tau(i)}$ is the set of sequences of node $i$'s neighbors in $\tau$, and $\neg i$ is the set of latent nodes except node $i$. Continuing with the two other approximations, we have

$$\log q^*(\tau|X) \stackrel{\pm}{=} \mathbb{E}_{q(\mathcal{B}, Z|X)} [\log p_\theta(X, Z, \mathcal{B}, \tau)]$$
$$\stackrel{\pm}{=} \mathbb{E}_{q(\mathcal{B}|X)} \left[ \sum_{\substack{(i,j) \in E(\tau) \\ Z_i, Z_j}} q(Z_i, Z_j|X) \log p_\theta(Z_i|Z_j, b(i, j)) \right] \tag{7}$$

and

$$\log q^*(b(i, j)|X) \stackrel{\pm}{=} \mathbb{E}_{q(\tau, \mathcal{B}_{\neg b(i,j)}, Z|X)} [\log p_\theta(X, Z, \mathcal{B}, \tau)]$$
$$\stackrel{\pm}{=} \mathbb{E}_{q(\tau|X)} \left[ \sum_{Z_i, Z_j} q(Z_i, Z_j|X) \log p_\theta(Z_i|Z_j, b(i, j), \tau) \right] + \log p_\theta(b(i, j)). \tag{8}$$

The exact computation of the expectations in Eq. (6) and (8) involves summing over all possible trees, which is computationally intractable. Luckily, we can easily approximate the expectation using importance sampling, and our new proposal distribution, $s_\phi(\tau)$, is outlined in the subsequent Sec. 3.2. We direct the reader to Appendix A for the derivations of the above variational distributions and the details of the algorithm.

To obtain good starting points for the variational distributions, we initialize $\phi$ with trees obtained using the Neighbor-Joining algorithm [14]. The details of the initialization are provided in Appendix A.2. Alg. 1 is a high-level algorithmic description of VaiPhy. The proof regarding the natural gradient is presented in Appendix I.

**Algorithm 1** VaiPhy in pseudocode – $\iota$ is the total number of iterations and $\eta$ is the natural gradient step size

```
 1: Input: X, φ, η, ι
 2: for iter = 1, . . . , ι do
 3:    Approximate log q*(Z|X) using s_φ(τ) {Eq. (6)}
 4:    q(Z_i) ← (1 − η)q(Z_i) + ηq*(Z_i)
 5:    φ_ij ← Σ_{α≠β} F_ij^{α,β}   {Eq. (5)}
 6:    q(τ|X) ← q*(τ|X)  {Eq. (7)}
 7:    q(B|X) ← q*(B|X) using s_φ(τ) {Eq. (8)}
 8: end for
 9: return q(τ, B, Z|X), φ
```

## 3.2 SLANTIS

We propose the method Sequential Look-Ahead Non-local Tree Importance Sampling (SLANTIS), which enables sampling of a tree, $\tau$, from a distribution, $s_\phi(\tau)$. SLANTIS has support over multifurcating trees with labeled internal vertices and is parameterized by $\phi$. To describe the algorithm, we next introduce some notation.

Using $\phi$, SLANTIS immediately constructs $W$, an $N \times N$ matrix describing all pairwise edge weights based on the substitution model,

$$W_{ij} = \sum_{\alpha,\beta:\alpha\neq\beta} \phi_{ij} \left( b(i,j) \, Q_{\alpha\beta} \right) + \sum_{\alpha,\beta:\alpha=\beta} \left( M - \phi_{ij} \right) \left( b(i,j) \, Q_{\alpha\beta} \right), \tag{9}$$

where $Q$ is the rate matrix of the JC69 model and $\alpha$ and $\beta$ are nucleotides. Moreover, let $\mathcal{G}$ be a graph with vertex set $\{N\}$ with edge weights specified by $W$. Let $\mathcal{G}_\mathcal{I} = \mathcal{G}\backslash\Lambda$, where $\Lambda$ is the $(V, E)$ of the leaves, i.e., the graph induced by the internal vertices of our trees. The pseudocode of the algorithm is presented in Appendix B. The main idea, which is borrowed from [6], is to order the edges of the graph and for each edge in the order, make a choice whether to include it in the sampled tree or not. The choice is made based on a Bernoulli trial, where the two probabilities are proportional to the weight of the maximum spanning tree (MST) containing the already selected edges and the currently considered edge and that of the MST containing the already selected edges but not the currently considered edge.[1] The action associated with the former outcome is to include the edge in the sampled tree, and the other outcome is not to include it.

To facilitate fast computation of the required MSTs, we first consider the edges of an MST $T_1$ of $\mathcal{G}$. After the edges of $T_1$, the edges of an MST $T_2$ of $\mathcal{G} \setminus T_1$ can be found. In general, $T_s$ is an MST of $\mathcal{G} \setminus \cup_{r=1}^{s-1} T_r$, and after the edges of $\cup_{r=1}^{s-1} T_r$, the edges of $T_s$ can be found. In the algorithm, we repeatedly take advantage of the property that for any tree $T$ and edge $e$ not in $T$, $T \cup e$ contains a unique cycle, and the removal of its heaviest edge different from $e$ produces the minimum weight subtree of $T \cup e$ containing $e$. If the considered edge is in $T$ and we choose to include it in the sampled tree, $\tau$, we recursively update $\log s_\phi(\tau)$ by adding the Bernoulli log probability corresponding to the edge. See Fig. 2 for an illustration.

## 3.3 The JC Sampler

The JC69 model is arguably one of the most frequently used substitution models in Bayesian phylogenetics. However, as far as we are aware, the model has only been used for obtaining maximum likelihood estimates of branch lengths. Here, we observe that the JC69 model can be seen as an i.i.d. Bernoulli likelihood over sites and leverage the Beta-Bernoulli conjugacy for obtaining a normalized distribution over branch lengths in variational inference. Below we show how to sample from this distribution and evaluate its likelihood using the change-of-variables formula.

---

[1]In [6], the candidate edges are selected from those that will make a *perfect matching*, and the probability of each edge is uniform.

Given a branch length $b(e) = b$, the probability of the $m$'th nucleotide *not* mutating along edge $e = (i, j)$ is, according to the JC69 model,

$$p = \left( \frac{1}{4} + \frac{3}{4} e^{-\frac{4}{3}b} \right)^{\mathbb{1}\{Z_i^m = Z_j^m\}}. \tag{10}$$

As all mutations are equiprobable for this model, the probability of the nucleotide mutating is thus $1 - p$, giving rise to the following distribution when all $M$ estimated mutations and non-mutations are observed and assuming mutations occur independently across sites

$$\text{Bern}\left(\phi_{ij}|p\right) = \prod_{m=1}^{M} \text{Bern}\left(\phi_{ij}^m|p\right) = \prod_{m=1}^{M} p^{1-\phi_{ij}^m}(1-p)^{\phi_{ij}^m} = p^{M-\phi_{ij}}(1-p)^{\phi_{ij}}, \tag{11}$$

where $\phi_{ij}^m$ is the expected number of mutations over the edge $(i, j)$ at site $m$. Assuming an uninformative Beta prior on the Bernoulli parameter, $p$, and recalling that Beta is a conjugate prior for the Bernoulli distribution, we get a posterior distribution over $p$, $\text{Beta}(p|M - \phi_{ij} + 1, \phi_{ij} + 1)$. Since we are ultimately interested in sampling $b$, not $p$, we find a transform, $f(\cdot)$, mapping $p$ to $b$, using Eq. (10);

$$f^{-1}(b) = \frac{1}{4} + \frac{3}{4} e^{-\frac{4}{3}b} = p \tag{12}$$

$$f(p) = -\frac{3}{4} \log \left( \frac{4}{3} \left( p - \frac{1}{4} \right) \right) = b. \tag{13}$$

In all, sampling branch lengths from the JC model include two straightforward steps, (i) draw $p$ from the Beta posterior distribution, and (ii) pass it through the transform, $b = f(p)$.

Of course, we are also interested in evaluating the likelihood of the sampled branch length. To do so, we use the change-of-variables formula; we let

$$s_\phi(b) = \text{Beta}(p|M - \phi_{ij} + 1, \phi_{ij} + 1) \left| \frac{df^{-1}(b)}{db} \right| \tag{14}$$

$$= \frac{p^{M-\phi_{ij}}(1-p)^{\phi_{ij}} e^{-\frac{4}{3}b}}{\text{B}(M - \phi_{ij} + 1, \phi_{ij} + 1)}, \tag{15}$$

where $\text{B}(\cdot)$ is the Beta function. This provides an efficient way of evaluating the likelihood of a branch length. In Alg. 2, we summarize the JC sampler with an algorithmic description.

---

**Algorithm 2** The JC Sampler

1: **Input:** $\phi, i, j$
2: Sample $p \sim \text{Beta}(p|M - \phi_{ij} + 1, \phi_{ij} + 1)$
3: Transform $b = f(p)$ {Eq. (13)}
4: Evaluate $s_\phi(b)$ {Eq. (14)}
5: **return** $b, s_\phi(b)$

---

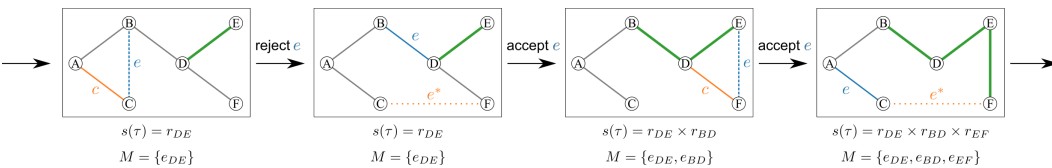

Figure 2: Propagation step of SLANTIS. The solid line is $\tau$ so far, and bold **green** lines are accepted edges. Dashed and dotted lines are the edges not in $\tau$. The blue color indicates the edge we consider ($e$), and the orange color indicates the alternative edge, either $c$ or $e^*$. $r_e = W(e)/(W(e) + W(c))$ or $r_e = W(e)/(W(e) + W(e^*))$. At the end of SLANTIS, the spanning tree with bold green lines is reported.

### 3.4 Assessing the Framework

The framework learns its parameters and distributions by maximizing the ELBO in Eq. (4). However, the IWELBO offers a tighter lower bound on the marginal log-likelihood. Using SLANTIS and the JC sampler, we can easily evaluate our framework using the IWELBO:

$$\mathcal{L}_L = \mathbb{E}_{\mathcal{B}_\ell, \tau_\ell \sim s_\phi(\mathcal{B}, \tau)} \left[ \log \frac{1}{L} \sum_{\ell=1}^{L} \frac{p_\theta(X, \mathcal{B}_\ell, \tau_\ell)}{s_\phi(\mathcal{B}_\ell, \tau_\ell)} \right], \tag{16}$$

where we may factorize the denominator as $s_\phi(\mathcal{B}, \tau) = s_\phi(\mathcal{B}|\tau)s_\phi(\tau)$, and $s_\phi(\mathcal{B}|\tau) = \prod_{e \in E(\tau)} s_\phi(b(e))$. Note that we have marginalized out the auxiliary variable, $Z$, in the target in Eq. (16), as we do not wish to compute $\mathcal{L}_L$ using our variational distributions.

## 4 Using VaiPhy to Parameterize the CSMC

As the tree topology space in A is considerably larger than the bifurcating $X$-tree space, obtaining competitive log-likelihood (LL) estimates is not straightforward. Namely, the uniform prior over the tree space in A, $p_\theta(\tau) = 1/(N - |X|)^{N-2}$, turns into a heavily penalizing term. When compared to baselines in smaller sample spaces, this will become a considerable negative factor for VaiPhy. Compare this to the prior when considering unrooted, bifurcating $X$-trees (which is the case for VBPI-NF): $p'_\theta(\tau) = 1/(2|X| - 5)!!$, where $|X|$ is the number of taxa. As an example, for $|X| = 10$ and $N = 18$, $p_\theta(\tau)$ distributes mass uniformly over 281 trillion trees, while the latter one merely considers 2 million trees.

Inspired by [26], we instead seek to use VaiPhy to parameterize a CSMC algorithm, thereby obtaining a procedure for projecting our trees to the lower-dimensional leaf-labeled tree space. Here, the prior is instead $p''_\theta(\tau) = 1/(2|X| - 3)!!$.

Since $\phi$ is not constrained to multifurcating trees, we can utilize the estimated number of mutations between vertices learned in A, also in the bifurcating $X$-tree space, $\Omega$. Here arises an interesting obstacle. In the CSMC algorithm, the internal vertices are *unlabeled*. Hence we are effectively blindfolded and need to estimate which entry in $\phi$ to use for a given edge. We achieve this by scoring edges based on how frequently the corresponding split of the leaf-set $X$ occurs in trees sampled from $s_\phi(\tau)$. See Appendix D.2 for more information.

At each rank $\rho \in [1, R]$, the CSMC proposes (i) a forest, $\mathcal{T}_\rho$, by *merging* the roots of two subtrees in $\mathcal{T}_{\rho-1}$, and (ii) a corresponding set of branch lengths, $\mathcal{B}_\rho$. This is done in parallel for $K$ *particles*, resampling particles with probability proportional to their importance weights, $w_\rho^k$, at every rank.[2] However, the reason to take advantage of a CSMC is that it allows us to estimate the marginal log-likelihood in the bifurcating $X$-tree space using the well-known formula,

$$\log p_\theta(X) \approx \log \prod_{\rho=1}^{R} \frac{1}{K} \sum_{k=1}^{K} w_\rho^k. \tag{17}$$

We let $\phi$ parameterize our two novel CSMC proposal distributions, $r_\phi(\mathcal{B}_\rho|\mathcal{B}_{\rho-1}, \mathcal{T}_\rho)$ and $r_\phi(\mathcal{T}_\rho|\mathcal{T}_{\rho-1})$. The former constructs a mixture of JC samplers and evaluates the proposal's likelihood in a multiple importance sampling fashion [7], while the latter forms a categorical distribution over the roots in $\mathcal{T}_{\rho-1}$, parameterized by a function of the likelihoods of trees sampled by SLANTIS. The in-detail descriptions can be found in Appendix D.

## 5 Experiments

### 5.1 Visualizing the JC Sampler

We conduct a toy experiment, investigating how the estimated number of mutations along an edge affects the behavior of the JC sampler. We would expect that a greater number of mutations should

---

[2]For more details and background of the CSMC algorithm, we point the reader to [26] and [33] for excellent descriptions.

generate a larger branch length. This is the case since the involved vertices are supposedly farther apart in terms of an evolutionary distance.

As demonstrated in Fig. 1, this advantageous behavior is indeed displayed. As $\phi_{ij}$ grows, the corresponding JC sampling distribution is shifted towards larger branch lengths. Interestingly, the JC sampler distributes its mass more symmetrically than the Log-Normal or Exponential distributions, which are typically used for modeling branch length distributions [33, 39, 37, 26].

It follows from Eq. (13) that branch lengths grow infinitely large as $p$ goes to 0.25. This phenomenon in phylogenetics is referred to as saturation. Saturation occurs for distantly related lineages and implies that the states of the nucleotides in these ancestral sequences are mutually independent. This is not an interesting case, hence the non-exhaustive parameter search of $\phi_{ij}$ in Fig. 1.

## 5.2 Density Estimation

Here we benchmark our methods, VaiPhy, and $\phi$-CSMC, in terms of LL estimates on seven real-world datasets, which we refer to as DS1-DS7 ([16, 13, 35, 17, 25, 40, 29]; in Appendix E, we provide additional information about the datasets). Additionally, to the extent possible, we have used the open-source code of our baselines to survey the benchmarks on these popular datasets. We only considered baselines that employ the JC69 model. Moreover, we assumed an Exp(10) prior over branch lengths and a uniform prior over tree topologies. This was aligned with the generative models used in all the original works of the baselines.

For all methods, we use the exact parameter configurations reported in the corresponding papers or specified in their available code. We let VBPI-NF [37] and VCSMC [26] train for the number of iterations as specified in their works. For MrBayes [28] with the SS method, we follow the protocol used in [39].

As we are benchmarking sophisticated versions of the phylogenetic inference CSMC [33], it seems relevant to include a vanilla CSMC. We define the vanilla CSMC as one that proposes branch lengths using the model prior and merges subtrees uniformly. There are no learnable parameters in the vanilla CSMC.

Indeed, there exist particle MCMC based approaches for parameterizing the CSMC, e.g., [33] or [34]; however, these have not previously been benchmarked on the datasets used here, and the latter uses the K2P substitution model. As such, we leave these comparisons for future work and constrain ourselves to VI based CSMCs.

Following [26], we give all CSMC methods $K = 2048$ particles. The parameters of VCSMC, VaiPhy, and $\phi$-CSMC are selected based on their best iteration, measured in LL. Using these parameters, we compute the LL estimates averaged over ten random seeds. We run VaiPhy and $\phi$-CSMC for 200 iterations and evaluate them on Eq. (16) and Eq. (17), respectively. Vanilla CSMC is also evaluated using Eq. (17).

In Table 1, we provide the mean LL scores and standard deviations. On all datasets except DS2, our $\phi$-CSMC is the superior CSMC method (highlighted in red). It also produces significantly smaller variations in its LL estimates compared to VCSMC and the Vanilla CSMC. On DS2, the Vanilla CSMC is surprisingly the winning CSMC.

Overall, MrBayes SS is the superior method on these datasets, followed by VBPI-NF. VaiPhy, despite its heavily penalizing prior distribution over tree topologies (see Sec. 4), still outperforms VCSMC and Vanilla CSMC on all datasets but DS2, with extremely low-variance estimates of the LL.

## 5.3 Wall-Clock Time and Memory Comparisons

Being a mean-field VI algorithm, VaiPhy truly enjoys the related benefits in terms of speed, which we demonstrate in this section. The experiments are performed on a high-performance computing cluster node with two Intel Xeon Gold 6130 CPUs with 16 CPU cores each. Each node in the cluster has 96 GiB RAM.

We run VaiPhy for 200 iterations, and 128 trees are sampled using SLANTIS to compute the expectations in Eq. (6) and (8). To compute $\mathcal{L}_L$, 3000 trees are sampled from SLANTIS after 200 iterations. The $\phi$-CSMC results include the aforementioned VaiPhy runs and 10 additional

Table 1: Real dataset LL estimates for the different methods. Our $\phi$-CSMC is, on average, the best-performing CSMC method (red). Overall, MrBayes SS achieves the highest LL estimates (**bold**). *Not available due to unresolved memory issues.

| DATA SET | (#TAXA, #SITES) | MRBAYES SS | VBPI-NF | VCSMC (JC) | VANILLA CSMC | VAIPHY | $\phi$-CSMC |
|---|---|---|---|---|---|---|---|
| DS1 | (27, 1949) | **-7032.45 ± 0.15** | -7108.40 ± 0.11 | -9180.34 ± 170.27 | -8306.76 ± 166.27 | -7490.54 ± 0 | -7290.36 ± 7.23 |
| DS2 | (29, 2520) | **-26363.85 ± 0.33** | -26367.70 ± 0.03 | -28700.7 ± 4892.67 | -27884.37 ± 226.60 | -31203.44 ± 3E$^{-12}$ | -30568.49 ± 31.34 |
| DS3 | (36, 1812) | **-33729.60 ± 0.74** | -33735.09 ± 0.05 | -37211.20 ± 397.97 | -35381.01 ± 218.18 | -33911.13 ± 7E$^{-12}$ | -33798.06 ± 6.62 |
| DS4 | (41, 1137) | **-13292.37 ± 1.42** | -13329.93 ± 0.11 | -17106.10 ± 362.74 | -15019.21 ± 100.61 | -13700.86 ± 0 | -13582.24 ± 35.08 |
| DS5 | (50, 378) | **-8192.96 ± 0.71** | -8214.61 ± 0.40 | -9449.65 ± 2578.58 | -8940.62 ± 46.44 | -8464.77 ± 0 | -8367.51 ± 8.87 |
| DS6 | (50, 1133) | **-6571.02 ± 0.81** | -6724.36 ± 0.37 | -9296.66 ± 2046.70 | -8029.51 ± 83.67 | -7157.84 ± 0 | -7013.83 ± 16.99 |
| DS7 | (64, 1008) | **-8438.78 ± 73.66** | -8650.49 ± 0.53 | N/A* | -11013.57 ± 113.49 | -9462.21 ± 1E$^{-12}$ | -9209.18 ± 18.03 |

independent CSMC runs with 2048 particles. VBPI-NF runs for 400,000 iterations and uses 100,000 trees to estimate the lower bound after training. Each experiment is repeated 10 times. The wall-clock time comparison of the methods on DS1 is presented in Fig. 3.

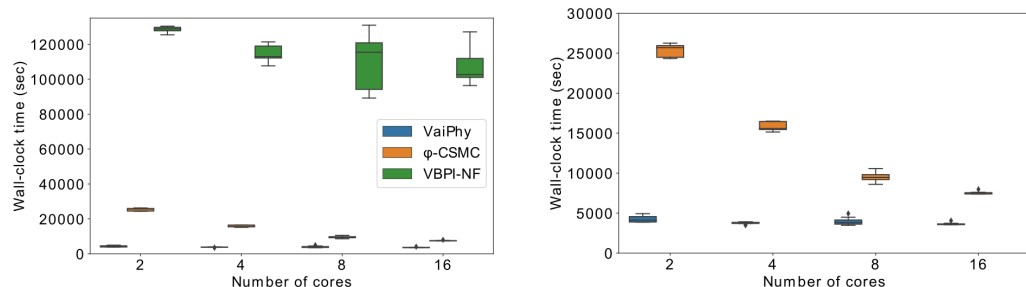

Figure 3: Wall-clock runtime results on DS1 data set with $\{2, 4, 8, 16\}$ cores. Left) Runtime results for VaiPhy, $\phi$-CSMC, and VBPI-NF. Right) A zoomed-in version of the top figure showing runtime results for VaiPhy and $\phi$-CSMC. Clearly, VaiPhy is consistently faster than its benchmarks.

We were not able to run VCSMC on less than 16 cores due to the method producing memory issues on DS1. However, for 16 cores, their runtime was around $6840$ seconds, with too small variations to show in a boxplot. VCSMC is thus faster than $\phi$-CSMC on 16 cores but still slower than VaiPhy.

VaiPhy is remarkably faster than other variational methods for each number of cores investigated. Although VBPI-NF produces impressive LL estimates, it is considerably slower than VaiPhy and the CSMC methods (all CSMCs are not shown in the figure). Additionally, using two cores, the wall-clock time required by VBPI-NF to arrive at VaiPhy's approximation accuracy was close to 11x that of VaiPhy on DS1. For completeness, we compared wall-clock times with MrBayes, although this is not appropriate given that the software has been highly optimized over decades and is written in C. And, indeed, MrBayes is the fastest algorithm when allowed multiple cores. However, for a single core on DS1, MrBayes ran for approximately 100 minutes while VaiPhy only needed an impressive 83 minutes on average for the same setup.

Finally, we discuss memory requirements. Our main variational baseline, VBPI-NF, used 8.4 million neural network weights and biases on DS1, while the number of learnable parameters (i.e., the dimensions of $\phi$ and $q(Z|X)$) in VaiPhy was 0.8 million; less than 10% of the former.

# 6 Conclusion

We have presented a modular framework for Bayesian phylogenetic inference, building on three novel methods: *VaiPhy*, *SLANTIS*, and the *JC sampler*. At the heart of our framework lies $\phi$, the estimated number of mutations matrix. Although $\phi$ was learned in a multifurcating tree space with labeled internal vertices, we showed that the evolutionary distances also served useful in the bifurcating $X$-tree space. We demonstrated this by using $\phi$ to parameterize two new proposal distributions for the CSMC, also introduced by us in this work; we referred to this new algorithm as $\phi$-CSMC. Our $\phi$-CSMC proved to be the best-performing CSMC in terms of marginal log-likelihood estimation on real datasets. Additionally, VaiPhy was up to *25 times faster* than its baselines on real data.

Speed is essential for making Bayesian phylogenetics a reality for future large data sets, in particular, single-cell data sets in medical investigations. Moreover, speed and the opportunity to leverage the VI methodology for phylogeny, as provided by our framework, i.e., our graph-based representation of the VI distribution over trees, $\phi$, opens up novel research opportunities. For instance, recently, ensemble and mixture models in VI have received a lot of attention [23, 27]. VaiPhy could obtain many ensemble components in parallel with low computational cost and thus has great potential to benefit from these findings. Moreover, more mixture components appear to imply significant improvements in terms of negative log-likelihood scores when learned by maximizing a multiple importance sampling-based objective, *MISELBO* [24]. This motivates researching how VaiPhy could be fitted using a MISELBO objective.

For many potential applications, models other than the JC69 model will be more suitable for the evolution that takes place over the edges of the phylogenetic tree. However, our JC sampler provides a great starting point. Reasonably, the JC sampler should serve as a better base distribution for normalizing flows than those currently employed [37]. Furthermore, in this work, the JC sampler samples the branch lengths independent of the tree topology. We expect that tree topology dependent branch sampling will enhance VaiPhy's performance. This will be investigated in our future work.

VI can facilitate Bayesian analysis of models that integrate phylogeny with other tasks that usually are performed as pre-processing or post-processing steps. The most important such example may very well be sequence alignment. Although Bayesian phylogenetic inference is currently considered state-of-the-art, this inference does not take the observed sequences associated with the taxa as input but a so-called multiple sequence alignment (MSA) of these, which is obtained by applying a heuristic to the sequences. This implies that the entire uncertainty related to the multitude of possible MSAs remains uncharacterized after a standard Bayesian phylogenetic inference, i.e., it does not affect the obtained posterior at all. For single-cell phylogenetic analysis of cancer cells, pre-processing may be even more involved. For instance, the so-called direct library preparation single-cell DNA data [36] for tumors require a pre-processing step in which the cells are clustered into cancer clones. For these clones, it is then desirable to perform a phylogenetic inference. Interestingly, the VI has already been launched for Bayesian clustering for this pre-processing step [30].

In conclusion, we have presented a VI algorithm for Bayesian phylogeny that, together with its building blocks, unlocks the potential of VI for fast Bayesian analysis in phylogeny and integrated phylogenetic analysis. The by now 25 year old methodology of MCMC based Bayesian phylogenetic analysis is certainly a mature and optimized methodology with great accuracy. However, VaiPhy is faster than MrBayes on a single core, and the underlying methodology is bound to lead to a sequence of increasingly faster and more accurate VI based phylogenetic software. Moreover, it paves the way for integrated VI based phylogenetic inference.

## Acknowledgments and Disclosure of Funding

First, we acknowledge the insightful comments provided by the reviewers, which have helped improve our work. This project was made possible through funding from the Swedish Foundation for Strategic Research grant BD15-0043, and from the Swedish Research Council grant 2018-05417_VR. The computations and data handling were enabled by resources provided by the Swedish National Infrastructure for Computing (SNIC), partially funded by the Swedish Research Council through grant agreement no. 2018-05973.

The authors declare no competing interests.

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
