# VaiPhy: a Variational Inference Based Algorithm for Phylogeny
# Appendix

## A  The VaiPhy Algorithm

### A.1  Update Equation Details

The update equations of VaiPhy follow the standard mean-field VI updates. The variational distribution factorizes over the tree topology, ancestral sequences of each latent vertex, and the branch lengths of all edges in the graph (Eq. (18));

$$q(\tau, \mathcal{B}, Z|X) = q(\mathcal{B}|X)q(\tau|X)q(Z|X) \tag{18}$$

$$= \prod_{e \in \mathcal{G}} q(b(e)|X)q(\tau|X) \prod_{i \in I(A)} q(Z_i|X). \tag{19}$$

$$\tag{20}$$

For brevity, we denote the set of neighbors of node $i$ with $N_\tau(i)$ and their sequences with $Y_{N_\tau(i)}$ in this section. Furthermore, $\neg i$ is the set of nodes except node $i$, and $C$ is a constant.

**Ancestral sequence update equation**

$$\log q^*(Z_i|X) \propto \mathbb{E}_{q(\tau,\mathcal{B},Z_{\neg i}|X)} \left[ \log p_\theta(X, Z, \mathcal{B}, \tau) \right] \tag{21}$$

$$= \sum_\tau \int_\mathcal{B} \sum_{Z_{\neg i}} q(\tau|X)q(\mathcal{B}|X)q(Z_{\neg i}|X) \left[ \log p_\theta(X, Z_i, Z_{\neg i}, \mathcal{B}, \tau) \right] d\mathcal{B} \tag{22}$$

$$= \sum_\tau \int_\mathcal{B} q(\tau|X)q(\mathcal{B}|X) \left[ \sum_{Y_j \in Y_{N_\tau(i)}} q(Y_j|X) \log p_\theta(Y_j|Z_i, b(i,j), \tau) \right] d\mathcal{B} + C \tag{23}$$

$$\stackrel{\pm}{=} \mathbb{E}_{q(\tau,\mathcal{B}|X)} \left[ \sum_{Y_j \in Y_{N_\tau(i)}} q(Y_j|X) \log p_\theta(Y_j|Z_i, b(i,j), \tau) \right] \tag{24}$$

$$\tag{25}$$

If $j$ is an observed node, $q(Y_j|X) = \mathbb{1}\{Y_j = X_j\}$.

**Tree topology update equation**

$$\log q^*(\tau|X) \propto \mathbb{E}_{q(\mathcal{B},Z|X)}\left[\log p_\theta(X,Z,\mathcal{B},\tau)\right] \tag{26}$$

$$= \sum_Z \int_{\mathcal{B}} q(\mathcal{B}|X)q(Z|X)\log p_\theta(X,Z,\mathcal{B},\tau)d\mathcal{B} \tag{27}$$

$$= \sum_Z \int_{\mathcal{B}} q(\mathcal{B}|X)q(Z|X)\log p_\theta(X,Z|\mathcal{B},\tau)d\mathcal{B} + \log p_\theta(\tau) + C \tag{28}$$

$$= \sum_{(i,j)\in E(\tau)} \sum_{Z_i,Z_j} \int_{b(i,j)} q(b(i,j)|X)q(Z_i,Z_j|X)\log p_\theta(Z_i|Z_j,b(i,j),\tau)db(i,j) \tag{29}$$

$$+ \log p_\theta(\tau) + C \tag{30}$$

$$\stackrel{\pm}{=} \mathbb{E}_{q(\mathcal{B}|X)}\left[\sum_{(i,j)\in E(\tau)} \sum_{Z_i,Z_j} q(Z_i,Z_j|X)\log p_\theta(Z_i|Z_j,b(i,j),\tau)\right] \tag{31}$$

$$\tag{32}$$

**Branch length update equation**

$$\log q^*(b(i,j)|X) \tag{33}$$

$$\propto \mathbb{E}_{q(\tau,\mathcal{B}_{\neg e(i,j)},Z|X)}\left[\log p_\theta(X,Z,\mathcal{B},\tau)\right] \tag{34}$$

$$= \sum_\tau \sum_Z \int_{\mathcal{B}_{\neg e(i,j)}} q(\tau|X)q(Z|X)q(\mathcal{B}_{\neg e(i,j)}|X)\left[\log p_\theta(X,Z,b(i,j),\mathcal{B}_{\neg e(i,j)},\tau)\right]d\mathcal{B}_{\neg e} \tag{35}$$

$$= \sum_\tau q(\tau|X) \sum_{Z_i,Z_j} q(Z_i,Z_j|X)\left[\log p_\theta(Z_i|Z_j,b(i,j),\tau)\right] + \log p_\theta(b(i,j)) + C \tag{36}$$

$$\stackrel{\pm}{=} \mathbb{E}_{q(\tau|X)}\left[\sum_{Z_i,Z_j} q(Z_i,Z_j|X)\log p_\theta(Z_i|Z_j,b(i,j),\tau)\right] + \log p_\theta(b(i,j)) \tag{37}$$

$$\tag{38}$$

In the experiments, we observed that using the branch length that maximizes the tree likelihood during optimization provided better results. Hence, during the training of VaiPhy, we used a maximum likelihood heuristic to update the branch lengths given a tree topology. After the training, we used the tree topologies sampled from SLANTIS and corresponding branch lengths sampled from the JC sampler to compute IWELBO.

## A.2 Neighbor-Joining Initialization

We utilize the NJ algorithm to initialize VaiPhy with a reasonable state. The sequence data is fed into BIONJ, an NJ algorithm, to create an initial reference phylogenetic tree using the PhyML software, version 3.3.20200621 [14, 15]. The branch lengths of the NJ tree are optimized with the same software. An example script to run PhyML is shown below.

```
phyml -i DS1.phylip -m JC69 --r_seed vbpi_seed -o l -c 1
```

The marginal likelihoods of internal vertices, $p_\theta(Z_i|X,\mathcal{B},\tau) \; \forall i \in I(A)$, are used to initialize the latent ancestral sequences, $q(Z_i|X)$. The optimized branch lengths are used as the initial set of lengths for $e \in E(\tau)$. The lengths of the edges that are not present in the NJ tree are initialized by computing the shortest path between the vertices using the Floyd-Warshall algorithm [10].

### A.3 Graphical Model

When training VaiPhy, we used the following branch length prior

$$p_\theta(\mathcal{B}|\tau) = \prod_{e \in E(\tau)} p_\theta(b(e)), \qquad (39)$$

where $p_\theta(b(e)) = \text{Exp}(10)$, and a uniform prior over tree topologies (in A):

$$p_\theta(\tau) = \frac{1}{N^{N-2}}. \qquad (40)$$

The prior when considering unrooted, bifurcating $X$-trees (which is the case for VBPI-NF):

$$p'(\tau) = \frac{1}{(2|X| - 3)!!}, \qquad (41)$$

and for rooted, bifurcating $X$-trees

$$p''(\tau) = \frac{1}{(2|X| - 5)!!}. \qquad (42)$$

## B SLANTIS

Here we provide two algorithmic descriptions of SLANTIS. Note that in Eq. (9), the weight matrix $W$ is in logarithmic scale. However, in Alg. 3 and 4, $W$ is in normal scale.

---

**Algorithm 3** SLANTIS in pseudocode

---

1: **Input:** $\phi, W$
2: $\mathcal{G} \leftarrow$ graph spanned by $W$
3: $\mathcal{G}_\mathcal{I} \leftarrow \mathcal{G} \setminus \Lambda$
4: Initialize $T_1$, e.g. by MST($W(\mathcal{G}_\mathcal{I})$)
5: Initialize $\tau \leftarrow T_1$, $s(\tau) \leftarrow 1$, $\mathcal{M} \leftarrow \{\}$ and $s \leftarrow 1$
6: **while** $|\mathcal{M}| < |V(\mathcal{G}_\mathcal{I})| - 1$ **do**
7: $\quad \mathcal{R} \leftarrow \mathcal{G}_\mathcal{I} \setminus T_s$
8: $\quad W_1 \leftarrow W(\mathcal{R})$
9: $\quad T_{s+1} \leftarrow$ MST($W_1$)
10: $\quad$ pass $T_s, T_{s+1}, \tau, W, \mathcal{M}$ to Algorithm 4
11: $\quad \tau, \mathcal{M}$ updated in Alg. 4
12: $\quad s(\tau) \leftarrow s(\tau) \times r$, from return value $r$ of Alg. 4
13: $\quad T_s \leftarrow T_{s+1}$, $s \leftarrow s + 1$
14: **end while**
15: Sample leaf connections
16: **for** $u \in V(\Lambda)$ **do**
17: $\quad i \leftarrow$ indexes of internal vertices in $\mathcal{G}$
18: $\quad p_1, ..., p_n \leftarrow \frac{W(u,i_1)}{\Sigma W(u,i)}, ..., \frac{W(u,i_n)}{\Sigma W(u,i)}$
19: $\quad v \leftarrow$ Categorical($p_1, ..., p_n$)
20: $\quad$ add $(u, v)$ to $\tau$
21: $\quad s(\tau) \leftarrow s(\tau) \times p_v$
22: **end for**
23: **return** $\tau$, $s(\tau)$

---

**Algorithm 4** SLANTIS - propagation

1: **Input:** $t_0, t_1, \tau, W, \mathcal{M}$
2: Initialize $r \leftarrow 1$
3: $I_1 \leftarrow sort(t_1)$, e.g. by sorting $W(t_1)$ in descending order and selecting the indices.
4: **for** $e \in t_0$ **do**
5:     **if** $e \in \tau$ **then**
6:         Set $\tau_0, \tau_1$ such that $e = cut(\tau_0, \tau_1)$
7:         **for** $e_1 \in I_1$ **do**
8:             **if** $e_1$ connects $\tau_0, \tau_1$ **then**
9:                 $e^* \leftarrow e_1$
10:                 **break** inner for loop
11:             **end if**
12:         **end for**
13:         $r_e = \frac{W(e)}{W(e) + W(e^*)}$
14:     **else if** $e \notin \tau$ **then**
15:         add $e$ to $\tau$ creating new graph $\tau^*$ with cycle $\mathcal{C}$
16:         $c \leftarrow \arg\min(W(\mathcal{C} \setminus \{\mathcal{M} \cup e^*\}))$
17:         **if** c = {} **then**
18:             **continue**
19:         **end if**
20:         remove $c$ from $\tau^*$
21:         $r_e = \frac{W(e)}{W(e) + W(c)}$
22:     **end if**
23:     $u \leftarrow \text{Uniform}(0, 1)$
24:     **if** $e \in \tau$ and $r_e < u$ **then**
25:         $\tau \leftarrow \tau_0 \cup e^* \cup \tau_1$
26:     **else if** $e \in \tau$ and $r_e \geq u$ **then**
27:         $M \leftarrow M \cup e$
28:         $r \leftarrow r \times r_e$
29:     **else if** $e \notin \tau$ and $r_e < u$ **then**
30:         **continue**
31:     **else if** $e \notin \tau$ and $r_e \geq u$ **then**
32:         $\tau \leftarrow \tau^*$
33:         $M \leftarrow M \cup e$
34:         $r \leftarrow r \times r_e$
35:     **end if**
36:     **if** $|\mathcal{M}| = |V(\mathcal{G}_\mathcal{I})| - 1$ **then**
37:         **break** for loop
38:     **end if**
39: **end for**
40: **return** $\tau, \mathcal{M}, r$

Fig. 4-7 show four different cases in SLANTIS propagation. In all of the figures, the left column is the current state of $\tau$, the middle column is two trees that are compared, and the right column is the selected tree. Solid lines indicate the edges in $\tau$, and bold green lines are accepted edges (edges in $M$). The blue color indicates the edge we consider ($e$) and orange color indicates the alternative edge; either $c$ or $e^*$. Depending on the configuration, the Bernoulli random variable is either $r_e = W(e)/(W(e) + W(c))$ or $r_e = W(e)/(W(e) + W(e^*))$. In order to add an edge to $M$, the edge must have blue color ($e \in \tau$) and must get accepted. At the end of the SLANTIS algorithm, the spanning tree with bold green edges, along with its sampled leaf connections, is returned.

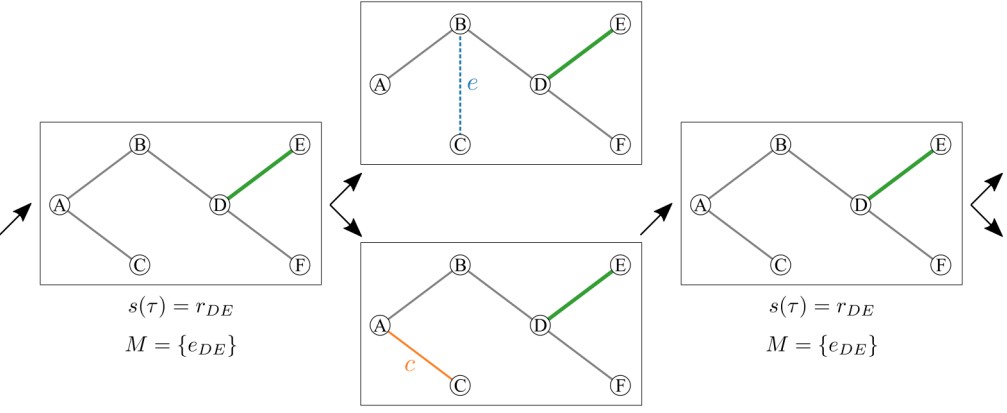

Figure 4: Propagation step of SLANTIS, a case where $e \notin \tau$ and $c \in \tau$. The Bernoulli r.v. is $r_e = W(e)/(W(e) + W(c))$. $e$ is rejected, therefore $\tau$, $s(\tau)$ and $M$ remain unchanged.

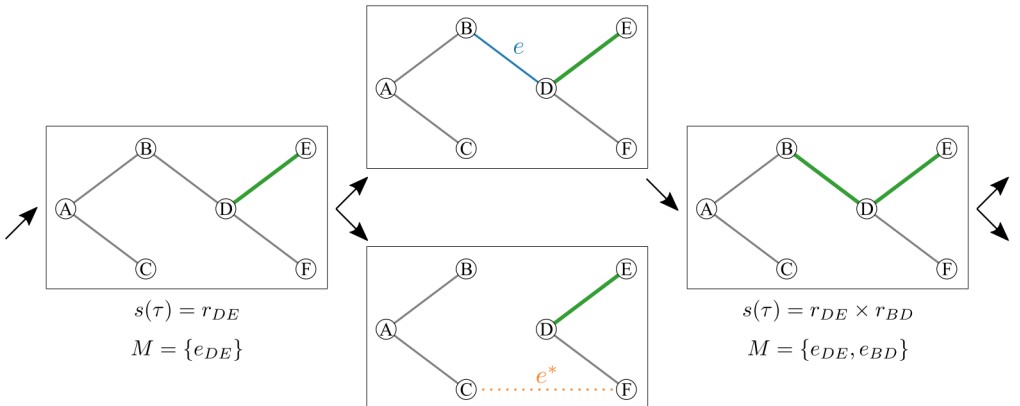

Figure 5: Propagation step of SLANTIS, a case where $e \in \tau$ and $e^* \notin \tau$. The Bernoulli r.v. is $r_e = W(e)/(W(e) + W(e^*))$. $e$ is accepted, therefore $\tau$, $s(\tau)$ and $M$ are updated.

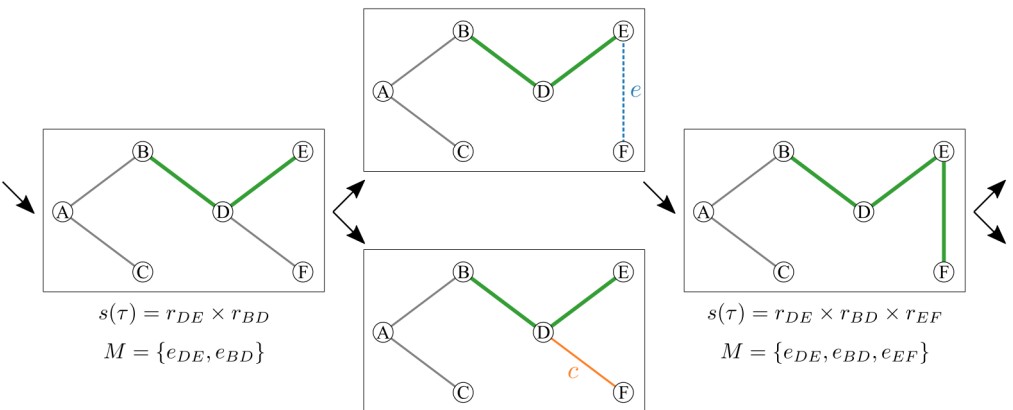

Figure 6: Propagation step of SLANTIS, a case where $e \notin \tau$ and $c \in \tau$. The Bernoulli r.v. is $r_e = W(e)/(W(e) + W(c))$. $e$ is accepted, therefore $\tau$, $s(\tau)$ and $M$ are updated.

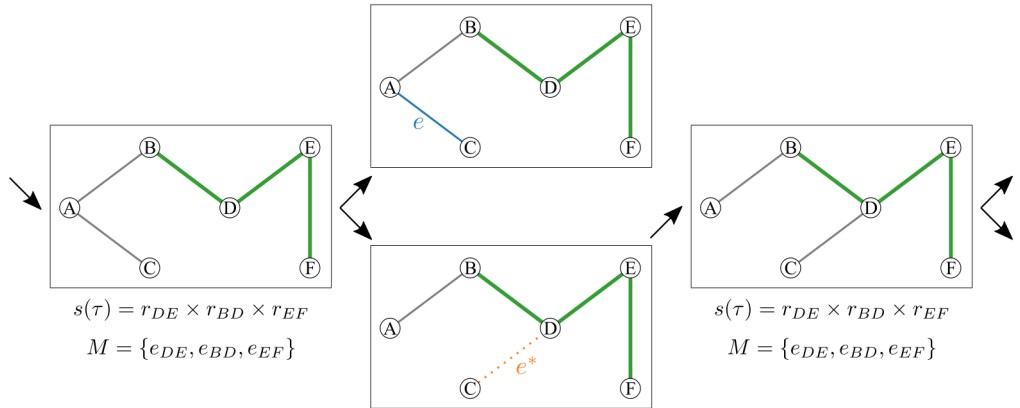

Figure 7: Propagation step of SLANTIS, a case where $e \in \tau$ and $e^* \notin \tau$. The Bernoulli r.v. is $r_e = W(e)/(W(e) + W(e^*))$. $e$ is rejected, therefore $\tau$, $s(\tau)$ and $M$ remain unchanged.

## C    The JC Sampler

In the JC Sampler, the probability of no substitutions is sampled from the Beta posterior distribution as follows;

$$p \sim \text{Beta}(p|M - \phi_{ij} + 1, \phi_{ij} + 1).$$

The Beta posterior distribution is parameterized by the Beta prior ($Beta(p|1, 1)$ in this case), the sequence length ($M$), and the expected number of substitutions between two nodes ($\phi_{ij}$).

Since the support of the Beta distribution is in $[0, 1]$, it is possible to sample $p < 0.25$. However, according to the JC69 model, such a case corresponds to saturation, and the logarithm in Eq. (13) becomes negative.

In practice, on the rare occasions where we encounter $p < 0.25$, we discard that sample and draw another $p$ from the Beta posterior distribution. Yet, the user can adjust the prior distribution parameters to change the behavior of the posterior distribution (e.g., skew the distribution away from $p < 0.25$).

## D    CMSC Proposal Distributions

Here we go into detail about our novel CSMC proposals.

### D.1    The Branch Length Proposal

Here we outline our new proposal distribution for sampling sets of branch lengths in the CSMC, $r_\phi(\mathcal{B}_\rho|\mathcal{B}_{\rho-1}, \mathcal{T}_\rho)$. In practice, sampling the $\mathcal{B}_\rho$ means that, given the two edges most recently added to $\mathcal{T}_\rho$ ($e$ and $e'$), we sample branch lengths using the JC sampler, $s_\phi(b(e))$, and concatenate $\{b(e), b(e')\}$ to the previous set, $\mathcal{B}_\rho = \{b(e), b(e')\} \cup \mathcal{B}_{\rho-1}$. The complicated part is identifying which entries in $\phi$ the edges correspond to.

We let the proposal distribution of branch lengths be a mixture of JC samplers while additionally allowing edge indices to be random variables:

$$r_\phi(b, b', e, e') = s_\phi(b|e)\, r(e|e')\, s_\phi(b'|e')\, r(e'). \tag{43}$$

Branch length $b$ given $e$ is sampled independently of $e'$, and vice versa; however, we assume that the edge indices are not mutually independent. If $e'$ is sampled first from the mixture, then $e$ is constrained to sharing the same parent node as $e$. We get the following priors

$$r(e') = \sum_{i,j \in \mathcal{C}'(e')} \frac{c(i,j)}{C} \delta_{i,j}(u, v), \tag{44}$$

where $c(i, j)$ is the number of occurrences of the edge in the pre-sampled SLANTIS trees, $C'$ is a normalizing constant, and $\delta_{i,j}(\cdot)$ is the Dirac delta with the mass on $i, j$. $\mathcal{C}(e')$ is the set of possible

labeled edge indices for $e'$. E.g., if the root in $e'$ is a leaf node, then we only need to consider edges involving that labeled leaf. To estimate the index of the unlabeled edge $e'$, we thus sample from $r(e')$. Similarly, for $r(e|e')$, the set $\mathcal{C}(e|e')$ is even smaller, as we have estimated the label of the parent.

The above results in the following likelihood function for the set of branch lengths:

$$r_\phi(\mathcal{B}_\rho|\mathcal{B}_{\rho-1}, \mathcal{T}_\rho) = \prod_{\bar{e} \in E(\mathcal{T}_\rho)} \sum_{i,j}^{N} s_\phi(b(\bar{e})) r_\phi(i,j|\bar{e}), \tag{45}$$

where $\bar{e}$ is the estimated edge index.

### D.2 The Merger Proposal

At each rank $\rho \in [1, R]$, the CSMC proposes a forest, $\mathcal{T}_\rho$, by *merging* the roots of two subtrees in $\mathcal{T}_{\rho-1}$.

Let $r_1, \ldots, r_R$ be the roots of a forest $\mathcal{T}_{\rho-1}$.[3] There are $\binom{R}{2}$ way to merge the roots. In the case of vanilla CSMC, the probability of merging roots $r_i$ and $r_j$ is uniform, $p(r_i, r_j) = 1/\binom{R}{2}$.

For $\phi$-CSMC, we sample $S$ trees from SLANTIS and corresponding branch lengths from JC sampler as a preprocessing step.[4] Let $X(r_i)$ and $X(r_j)$ be the set of observations in the subtrees with roots $r_i$ and $r_j$ respectively. We define a function $O(X(r_i), X(r_j), \tau)$, that checks each edge in $\tau$ and returns 0 if there is an edge that separates the observations into two disjoint sets such that one set of observations is $X(r_i) \cup X(r_j)$. The probability of merging subtrees with roots $r_i$ and $r_j$ is:

$$p(r_i, r_j) \propto \sum_{s=1}^{S} p_\theta(X|\tau^s, \mathcal{B}^s) \mathbb{1}\{O(X(r_i), X(r_j), \tau^s) = 0\} \tag{46}$$

## E  Biological Data Sets

The details of the biological data sets are presented in Table 2. The former and up-to-date TreeBASE Matrix IDs are displayed [31].

Table 2: Biological data set details

| Name | TreeBASE Matrix ID Legacy | Up-to-date Legacy | Taxa | Sites | Reference |
|------|--------|----------------|------|-------|-----------|
| DS1 | M336 | M2017 | 27 | 1949 | [16] |
| DS2 | M501 | M2131 | 29 | 2520 | [13] |
| DS3 | M1510 | M127 | 36 | 1812 | [35] |
| DS4 | M1366 | M487 | 41 | 1137 | [17] |
| DS5 | M3475 | M2907 | 50 | 378 | [25] |
| DS6 | M1044 | M220 | 50 | 1133 | [40] |
| DS7 | M755 | M2261 | 64 | 1008 | [29] |

---

[3]For brevity, we focus on a single particle.
[4]For the experiments, we set S=3,000.

## F MrBayes Experiment

Following the experimental setup in [37], we ran MrBayes version 3.2.7 [28] (written in C) with the stepping-stone algorithm and summarized the results of 10 independent runs. We set the number of generations to 10,000,000 and the number of chains per run to 4. The mean and the standard deviation of the marginal log-likelihood estimates are calculated by using the results reported in MrBayes output file. An example script to run MrBayes is shown below.

```
BEGIN MRBAYES;
set autoclose=yes nowarn=yes Seed=123 Swapseed=123;
lset nst=1;
prset statefreqpr=fixed(equal);
ss ngen = 10000000 nruns=10 nchains=4 printfreq=1000 \
samplefreq=100 savebrlens=yes filename=mrbayes_ss_out.txt;
END;
```

## G VBPI-NF Experiment

We used the codes in the author's GitHub repository, which is based on PyTorch, to reproduce their results on the biological data sets. VBPI-NF requires precomputed bootstrap trees to gather the support of conditional probability tables. For DS[1-4], we used the bootstrap trees provided by the authors in their code repository. For DS[5-7], following the guidelines in [37], UFBoot is used to create 10 replicates of 10,000 bootstrap trees [18]. An example script is shown below.

```
iqtree -s DS5 -bb 10000 -wbt -m JC69 -redo
```

We chose RealNVP(10) variant for comparison. To reproduce the results on biological datasets, we used RealNVP flow type with 10 layers. The step size for branch length parameters is set to 0.0001.[5] We introduced a seed parameter in their code to be able to reproduce the runs. We used default settings for the rest of the parameters. An example script is shown below.

```
python main.py --dataset DS5 --flow_type realnvp --Lnf 10 \
--stepszBranch 0.0001 --vbpi_seed 1
```

## H VCSMC Experiment

We attempted to reproduce the JC model results of VCSMC [26] using the authors' GitHub repository (based on TensorFlow 1) and default hyperparameters: 100 epochs, batch size 256, learning rate 0.001, branch length initialization $\ln(10)$. Similar to the VBPI-NF experiments, we added a seed argument for reproducibility.

```
python runner.py --dataset hohna_data_1 --jcmodel True --num_epoch 100 \
--n_particles 2048 --batch_size 256 --learning_rate 0.001 --nested False --seed 1
```

After running 100 epochs, the parameters $\lambda_{max}$ and $Q_{max}$ (although $Q_{max}$ is fixed for the JC model) of the epoch with maximum $\log \hat{p}(X)$ were selected. Then CSMC was run ten times using $\lambda_{max}$ and $Q_{max}$. The results in Table 1 correspond to the means and standard deviations for $\log \hat{p}(X)$ and $p(X|\tau, B)$ of these ten evaluation runs. In the VCSMC paper, the results are obtained by alternating optimization (updating $\lambda$ and $Q$) and inference (estimating $p(X)$ via the approximated ELBO, $\hat{\mathcal{L}}_i$) in 100 iterations, ultimately reporting $\hat{\mathcal{L}}^\star = \max_i \{\hat{\mathcal{L}}_i\}_{i=1}^{100}$. Unfortunately, $\hat{\mathcal{L}}^\star$ will have a bias proportional to the variance of the selected $\hat{\mathcal{L}}_i$. Hence, $\mathbb{E}[\hat{\mathcal{L}}^\star]$ is not necessarily an ELBO. In practice, VCSMC produced $\hat{\mathcal{L}}_i$ with high variance (see Table 1), making it unclear if $\hat{\mathcal{L}}^\star$ results from a $\hat{\mathcal{L}}_i$ with a high mean or the variance. Instead, finding $\hat{\mathcal{L}}^\star$ and then rerunning the inference with the corresponding parameters for different seeds produces a fair benchmark; this is what we report.

The VNCSMC code in the authors' GitHub repository gave rise to non-trivial memory issues that we couldn't resolve; hence, the VNCSMC method is excluded from this paper.

---

[5]We consulted the authors.

# I Proof of Convex Combination for Natural Gradient for Exponential Families

The proof is two-folded: **Part 1**, where we prove that the Natural Gradient based VI update equation for Exponential Families (EF) is computed as a convex combination of old and new VI update equations; **Part 2**, where we prove that the VaiPhy distributions that should belong to EF, belong to EF.

**Part 1**: We basically prove what was stated in stochastic variational inference (SVI) [19] using SVI notation. Let $\mathcal{L}(q(\beta))$ denote the ELBO as a function of the variational distribution of the interest, as the goal is to maximize ELBO w.r.t. $q(\beta)$. We know that

$$\mathcal{L}(q(\beta)) = \mathbb{E}_q[\log p(\beta, x, Z, \alpha)] - \mathbb{E}_{q(\beta)}[\log q(\beta)] \tag{47}$$

$$= \mathbb{E}_q[\log p(\beta|x, Z, \alpha) + \log p(x, Z, \alpha)] - \mathbb{E}_{q(\beta)}[\log q(\beta)] \tag{48}$$

$$\overset{\pm}{=} \mathbb{E}_q[\log p(\beta|x, Z, \alpha)] - \mathbb{E}_{q(\beta)}[\log q(\beta)]. \tag{49}$$

Note that for readability, [19] uses $q(\beta)$ instead of $q(\beta|\lambda)$, where $\lambda$ is the natural parameter. They assume that $q(\beta)$ and $p(\beta|x, Z, \alpha)$ belong to the same exponential family. However, we prove that they can merely belong to EF. Therefore, we have the following:

$$q(\beta) = H(\beta)e^{\lambda^T T(\beta) - A_g(\lambda)}, \tag{50}$$

$$p(\beta|x, Z, \alpha) = H(\beta)e^{\eta_g(x, Z, \alpha)^T T(\beta) - A_g(\eta_g(x, Z, \alpha))}, \tag{51}$$

$$\begin{aligned}
\mathcal{L}(q(\beta)) &= \mathbb{E}_{-q(\beta)}\left[\mathbb{E}_{q(\beta)}[\log H(\beta) + \eta_g(x, Z, \alpha)^T T(\beta) - A_g(\eta_g(x, Z, \alpha))]\right] \\
&\quad - \mathbb{E}_{q(\beta)}[\log H(\beta) + \lambda^T T(\beta) - A_g(\lambda)] \\
&\overset{\pm}{=} \mathbb{E}_{-q(\beta)}\left[\mathbb{E}_{q(\beta)}[\eta_g(x, Z, \alpha)^T T(\beta) - A_g(\eta_g(x, Z, \alpha))]\right] - \mathbb{E}_{q(\beta)}[\lambda^T T(\beta) - A_g(\lambda)] \\
&\overset{\pm}{=} \mathbb{E}_{-q(\beta)}\left[\mathbb{E}_{q(\beta)}[\eta_g(x, Z, \alpha)^T T(\beta)]\right] - \mathbb{E}_{q(\beta)}[\lambda^T T(\beta) - A_g(\lambda)] \\
&= \mathbb{E}_{-q(\beta)}\left[\eta_g(x, Z, \alpha)^T \mathbb{E}_{q(\beta)}[T(\beta)]\right] - \lambda^T \mathbb{E}_{q(\beta)}[T(\beta)] + A_g(\lambda).
\end{aligned} \tag{52}$$

The final line above corresponds to Eq. 13 in [19], except that they changed the measure so that they got $\mathbb{E}_\lambda[T(\beta)]$ instead of $\mathbb{E}_{q(\beta)}[T(\beta)]$; see the proof below.

**Lemma I.1** *It holds that* $\nabla_\lambda A_g(\lambda) = \mathbb{E}_{q(\beta)}[T(\beta)]$.

*Proof.*

Since the derivative of the logarithm is the inverse of the variable, we have

$$\nabla_\lambda A_g(\lambda) = \nabla_\lambda \left\{ \log \int H(\beta) \exp\{\lambda^T T(\beta)\} d\beta \right\} = \frac{\nabla_\lambda \int H(\beta) \exp\{\lambda^T T(\beta)\} d\beta}{\int H(\beta) \exp\{\lambda^T T(\beta)\} d\beta}. \tag{53}$$

Taking the derivative of the numerator in the above, we get

$$\frac{\int H(\beta) T(\beta) \exp\{\lambda^T T(\beta)\} d\beta}{\int H(\beta) \exp\{\lambda^T T(\beta)\} d\beta}. \tag{54}$$

We now rewrite the above and then multiply and divide the expression by $\exp\{A_g(\lambda)\}$, which results in

$$\frac{\int H(\beta)T(\beta)\exp\{\lambda^T T(\beta)\}d\beta}{\int H(\beta)\exp\{\lambda^T T(\beta)\}d\beta} = \int T(\beta)\frac{H(\beta)\exp\{\lambda^T T(\beta)\}d\beta}{\int H(\beta)\exp\{\lambda^T T(\beta)\}d\beta} \tag{55}$$

$$= \int T(\beta)\frac{H(\beta)\exp\{\lambda^T T(\beta)\}}{\exp\{A_g(\lambda)\}}\frac{\exp\{A_g(\lambda)\}}{\int H(\beta)\exp\{\lambda^T T(\beta)\}d\beta}d\beta. \tag{56}$$

We know that exponentiation of the log-normalizer gives us $\exp\{A_g(\lambda)\} = \int \exp\{\lambda^T T(\beta)\}H(\beta)d\beta$. Therefore, we have

$$\int T(\beta)\frac{H(\beta)\exp\{\lambda^T T(\beta)\}}{\int \exp\{\lambda^T T(\beta)\}H(\beta)d\beta}\frac{\int \exp\{\lambda^T T(\beta)\}H(\beta)d\beta}{\int H(\beta)\exp\{\lambda^T T(\beta)\}d\beta}d\beta = \tag{57}$$

$$\int T(\beta)\frac{H(\beta)\exp\{\lambda^T T(\beta)\}}{\int \exp\{\lambda^T T(\beta)\}H(\beta)d\beta}d\beta = \int T(\beta)q(\beta)d\beta = \mathbb{E}_{q(\beta)}[T(\beta)]. \tag{58}$$

Now that we have proved **Lemma H.1**, we know that $\mathbb{E}_{q(\beta)}[T(\beta)] = \nabla_\lambda A_g(\lambda)$; thus, we can write the ELBO as

$$\mathcal{L}(\lambda) = \mathbb{E}_{-q(\beta)}\big[\eta_g(x, Z, \alpha)^T \mathbb{E}_{q(\beta)}[T(\beta)]\big] - \lambda^T \mathbb{E}_{q(\beta)}[T(\beta)] - A_g(\lambda)$$
$$= \nabla_\lambda A_g(\lambda)\Big(\mathbb{E}_{-q(\beta)}[\eta_g(x, Z, \alpha)] - \lambda\Big) - A_g(\lambda). \tag{59}$$

Taking the derivative of the ELBO w.r.t. $\lambda$, we get the

$$\nabla_\lambda \mathcal{L}(\lambda) = \nabla_\lambda^2 A_g(\lambda)\Big(\mathbb{E}_{-q(\beta)}[\eta_g(x, Z, \alpha)] - \lambda\Big) + \nabla_\lambda A_g(\lambda)\Big(-1\Big) + \nabla_\lambda A_g(\lambda)$$
$$= \nabla_\lambda^2 A_g(\lambda)\Big(\mathbb{E}_{-q(\beta)}[\eta_g(x, Z, \alpha)] - \lambda\Big). \tag{60}$$

Knowing that $\nabla_\lambda^2 A_g(\lambda) = FIM(\lambda)$,[6] if we put the above gradient into the formula of the natural gradient, we get

$$\lambda^{t+1} = \lambda^t + \gamma\Big(FIM^{-1}(\lambda)\nabla_\lambda \mathcal{L}(\lambda)\Big)$$
$$= \lambda^t + \gamma\Big(\mathbb{E}_{-q(\beta)}[\eta_g(x, Z, \alpha)] - \lambda^t\Big) \tag{61}$$
$$= (1 - \gamma)\lambda^t + \gamma\mathbb{E}_{-q(\beta)}[\eta_g(x, Z, \alpha)].$$

**Part 2**: We prove that **Part 1** holds for VaiPhy, i.e., we prove the two corresponding distributions in VaiPhy's ELBO follow EF. That is $q(Z|X)$ and $p(Z, X|\mathcal{B}, \tau)$. Note that if the latter probability belongs to EF, it implies that $p(Z|X, \mathcal{B}, \tau)$ and $p(X|\mathcal{B}, \tau)$ belong to EF since we know that the likelihood $p(X|\mathcal{B}, \tau)$ is a JC model and itself follows EF; more details of JC being formulated as EF will be provided in the rest of this section.

In the normal scale, the update equation for latent nodes in the tree is

$$q^*(Z_u|X) \propto \prod_{\tau, \mathcal{B}}\Bigg[p_\theta(Z_r|\tau)^{I(Z_u = Z_r)} \times p_\theta(Z_u|Y_{pa(u)}, \mathcal{B}, \tau)^{q(Y_{pa(u)}|X)}$$
$$\times \prod_{Y_w \in Y_{C_\tau(u)}} p_\theta(Y_w|Z_u, \mathcal{B}, \tau)^{q(Y_w|X)}\Bigg]^{q(\tau, \mathcal{B}|X)}. \tag{62}$$

---

[6]Fisher Information Matrix.

We now prove that $q(Z|X)$ follows EF. In Eq. 62, each term involves $p$ with some exponent value calculated by either the previous iteration or the current one—we refer to the exponent as $\alpha$. Each term can be written as a canonical form of EF: $1 \times \exp\{\alpha \log p - 0\}$. Now, $q(Z_u|X)$ is the product over EF distributions; hence $q(Z_u|X)$ also belongs to EF, i.e., the canonical product of two EF is

$$h(x)e^{\theta_1 T(x) - A(\theta_1)} \times h(x)e^{\theta_2 T(x) - A(\theta_2)} = h(\hat{x})e^{(\theta_1 + \theta_2)T(x) - \hat{A}(\theta_1, \theta_2)}, \tag{63}$$

where $\theta$'s are the natural parameters.

Moreover, in Eq. 62, each $p$, except the probability of the root which is constant, i.e., $0.25^M$, follows the JC model's distribution, which is defined by the following, in which up to a constant, belongs to EF.

$$p(Z_u = i | Z_w = j, \tau, \mathcal{B}) = \frac{1}{4} + \frac{3}{4}e^{-\frac{4}{3}\mathcal{B}_{uw}} \quad \text{if} \quad i = j$$

$$p(Z_u = i | Z_w = j, \tau, \mathcal{B}) = \frac{1}{4} - \frac{1}{4}e^{-\frac{4}{3}\mathcal{B}_{uw}} \quad \text{if} \quad i \neq j$$

To prove that the above belongs to EF, one can rewrite $p(Z_u = i | Z_w = j, \tau, \mathcal{B}) \overset{\pm}{=} \frac{3}{4}e^{-\frac{4}{3}\mathcal{B}_{uw}}$, where $h(.) = \frac{3}{4}$, $T(.) = -\frac{4}{3}$, and $A(.) = 0$ in the EF's canonical form.

Next, we prove that $p(Z, X | \mathcal{B}, \tau)$ follows EF. We can refer to all vertices as $Y$, and we again assume the root vertex has a fixed probability that is $0.25^M$; therefore, we can write: $p(Y | \mathcal{B}, \tau) = \prod_{i \in V(\tau), i-1 = Pa(i)} p(Y_i | Y_{i-1}) p(Y_r | \tau) \propto \prod_{i \in V(\tau), i-1 = Pa(i)} p(Y_i | Y_{i-1})$. Now each $p(Y_i | Y_{i-1})$ follows a JC model, and in the previous proof, we showed that JC follows EF. Thus, the product of JC distributions also follows EF, again proven in the previous proof.