# OpenReview forum: "VaiPhy: a Variational Inference Based Algorithm for Phylogeny"
_NeurIPS.cc/2022/Conference — NeurIPS 2022 Accept_

### Official Review · Reviewer_aBtL · 2022-07-01

**Rating:** 8
**Confidence:** 3
**Soundness:** 3 good
**Presentation:** 3 good
**Contribution:** 4 excellent

**Summary:**

The paper introduces two methods for inference of phylogenetic trees from sequence data. Taking a variational inference approach, the VaiPhy method is constructed using two new samplers: SLANTIS and JC. The ability to estimate the expected number of mutations over edges leads to a variant of the CSMC method denoted \phi-CSMC. The methods are experimentally tested and compared to state-of-the-art methods on 7 datasets.

**Questions:**

* The sentence "VaiPhy is remarkably much faster than all other methods for all numbers of cores (except for
278 MrBayes in special cases, see further down)" seems a little strong when it is followed by "MrBayes is the fastest algorithm when allowed multiple cores." Can the first sentence can be expressed more clearly?

**Limitations:**

yes

**Strengths And Weaknesses:**

Strengths:
* introduction of several new techniques for variational inference in phylogenetic analysis
* important, high impact application area
* well-presented and technically sound paper (as far as I can evaluate)
* while the two methods only outperform existing methods in some cases, the techniques show a path to future advances with potential to significantly improve the state-of-the-art

Weaknesses:
* variational inference has been used before for phylogenetic analysis, so in that sense the underlying idea is not fundamentally new. However, as far as I can judge, the proposed methods still presents significant novelty and originality

---

> ### Author Response · Authors · 2022-08-02
> **Response to Reviewer aBtL**
>
>
> ### Weaknesses - Novelty of VI in Phylogenetics:
> > variational inference has been used before for phylogenetic analysis, so in that sense the underlying idea is not fundamentally new. However, as far as I can judge, the proposed methods still presents significant novelty and originality
>
> Indeed VI has been used for phylogenetic tree inference; however, please note that there is a huge difference between VBPI-NF, which has a variational distribution over a fixed set of tree topologies that is optimized using an ELBO, and VCSMC which uses an SMC to define a distribution over trees. For this reason, we consider VBPI-NF to be a VI method, and VCSMC to be a CSMC method. So, VaiPhy is the first VI method that is not constrained to a given set of tree topologies. Also, VaiPhy is the first method to use CAVI update equations for phylogenetic tree inference. Additionally, as pointed out by Reviewer vFFt, sampling branch lengths directly from the JC69 model is novel, and SLANTIS is a new algorithm for sampling multifurcating trees.
>
> ### Questions - MrBayes Runtime Comparison
> > The sentence "VaiPhy is remarkably much faster than all other methods for all numbers of cores (except for 278 MrBayes in special cases, see further down)" seems a little strong when it is followed by "MrBayes is the fastest algorithm when allowed multiple cores." Can the first sentence can be expressed more clearly?
>
> We updated line 277 as follows; "VaiPhy is remarkably faster than other variational methods for each number of cores investigated." and kept the statement in lines 286-287 as is.

---

### Official Review · Reviewer_Mtqk · 2022-07-05

**Rating:** 6
**Confidence:** 4
**Soundness:** 3 good
**Presentation:** 3 good
**Contribution:** 3 good

**Summary:**

The authors infer the posterior distribution over phylogenetic trees given only the leaves of the tree. To do so, the authors propose a method composed of three elements. A first one (VaiPhy) is a mean-field VI algorithm that returns the posterior distribution of the tree, internal vertices and set of branch lengths. VaiPhy uses the second element (JC sampler) to sample branch lengths and the third element (SLANTIS) to sample trees.  SLANTIS enables them to sample the branches of the tree one after the other, with probabilities proportional to the total weight of the maximum spanning tree. The JC sampler gives access to a normalised distribution over branch lengths, using the Jukes-Cantor model.

The authors compare VaiPhy to several other models, baselines and a VaiPhy-parametrised CSCM, using the IWELBO as metrics and wall-clock runtime. The authors show that among their proposed variants, VaiPhy-parametrised CSCM give the best results. While it is not better than MrBayes or VBNI-NF, it is significantly faster than VBNI-NF.


**Questions:**

I believe that the paper is overall well-written and structured. I have nevertheless a few clarification questions concerning the method and questions regarding the benchmark. Additionally, the method MrBayes SS seems to perform better than the proposed approach. Therefore, I think that the work is not ready for publication at NeurIPS in its current form but I would be happy to increase the score if the following questions were answered.

*****



### VaiPhy
One step in VaiPhy is to compute the expected number of mutations of an edge. The authors claim that the maximum number of mutations between two vertices is the number of sites. I believe that it is possible to have unobserved mutations, for example when the time step between two vertices is large compared to the mutation rate. I would be interested to know if this is something the authors are accounting for, and how or why not if it is not taken into account.

VaiPhy seems to consider that the nucleotides are independent. However, it is possible that some mutations co-occur or are mutually exclusive. Could these first order interactions be taken into account in VaiPhy?

### SLANTIS
The separation between previous work and proposed approach is unclear for SLANTIS. The authors could perhaps make a better distinction between the two in the document.

Due to the probabilistic nature of SLANTIS, is it possible that the leaves are not always all included in the reconstructed tree? If yes, how does this affect the performance?

### Parameterized CSMC
I believe that using VaiPhy to parameterize CSMC instead of using directly VaiPhy is impactful for a small number of taxa but could be less so for larger number of taxa. Have the authors compared the two models when the number of taxa is larger than 100?

### Experiments
I think that the experiments are interesting but I would like to better understand how the proposed approach performs and scales in terms of number of taxa or sites for example compared to the baselines.

Figure 1, which compares the number of mutations with branch length, is an interesting plot. I was wondering if it would be possible to also evaluate the reconstructed tree. This could be done on simulated data where the tree (and sequences of the ancestors) are known but only the leaves are used for phylogeny reconstruction.




**Minor comments**
- Line 4: change the word formidable for something less positive
- Line 14: "the, to the best" -> phrasing to change
- Line 22, 102: "phylogentic" -> phylogenetic
- Line 76: define CSMC
- Equation 6: define the subscript -i
- Equation 7: the right hand side is missing the dependency in T
- Equation 9: it is worth repeating what $Q_{\alpha\beta}$ is
- line 146: Algorithm B -> algorithm 3
- The sentence line 153-154 is very unclear, it is worth reformulating it.

---------
In light of the authors' response to the reviewers, I am in favor of accepting the paper and increase the grade to 6.



**Limitations:**

I think that the authors could better address the limitations of their work in a few sentences.

**Strengths And Weaknesses:**

### Originality
The methods proposed by the authors is novel as it combines and adapts several techniques to a specific problem, which is phylogeny inference in single-cell data.

### Significance
Phylogeny inference is an important problem and is in need of approaches that scale to the size of current single-cell datasets for example. The authors do a step in this direction. However, MrBayes SS seems to both perform better and is faster than the proposed approach.

### Clarity
The paper is overall well-written, apart from a few sentences / typos that are mentioned at the next section.

### Quality
While the overall quality of the paper is good, including maths and structure of the paper, I think that some points need to be better explained. They are the subject of the next section.

---

> ### Author Response · Authors · 2022-08-02
> **Response to Reviewer Mtqk - Part 1/2**
>
> ### Weaknesses - Significance
> > Phylogeny inference is an important problem and is in need of approaches that scale to the size of current single-cell datasets for example. The authors do a step in this direction. However, MrBayes SS seems to both perform better and is faster than the proposed approach.
>
> The main goal of VI in phylogenetics is for sure to surpass MrBayes with respect to both of these metrics. However, as pointed out in the first paragraph of Reviewer vFFt's limitations section and in the discussion of our paper, this is a work in progress. Meanwhile, the brilliant MrBayes framework has been developed and fine-tuned over decades.
> Furthermore, we strongly believe that this VI approach to phylogeny may be extended to more complex models; for instance, models using single-cell data in the cancer context. We currently mention this in the conclusion, but will elaborate on this in the final version by including the following text:
> "In particular, VI can facilitate Bayesian analysis of models that integrate phylogeny with other tasks that usually are performed as pre-processing or post-processing steps. The most important such example may very well be sequence alignment. Although, Bayesian phylogentic inference currently is  considered state-of-the-art, this inference does not take the observed sequences, associated with the taxa, as input but a so-called multiple sequence alignment (MSA) of these, which is obtained by applying a heuristic to the sequences. This implies that the entire uncertainty related to the multitude of possible MSAs remains uncharacterised after a standard Bayesian phylogenetic inference, i.e., it does not affect the obtained posterior at all. For single-cell phylogentic analysis of cancer cells the pre-processing may be even more involved. For instance, so-called Direct Library Preparation (DLP) single-cell DNA data for tumors require a pre-processing step in which the cells are clustered into cancer clones. For these clones it is then desirable to perform a phylogenetic  inference. Interestingly, for this pre-processing step, the VI methodology has already been launched for Bayesian clustering."
>
> ### Questions - VaiPhy
> > One step in VaiPhy is to compute the expected number of mutations of an edge. The authors claim that the maximum number of mutations between two vertices is the number of sites. I believe that it is possible to have unobserved mutations, for example when the time step between two vertices is large compared to the mutation rate. I would be interested to know if this is something the authors are accounting for, and how or why not if it is not taken into account.
> VaiPhy seems to consider that the nucleotides are independent. However, it is possible that some mutations co-occur or are mutually exclusive. Could these first order interactions be taken into account in VaiPhy?
>
> This is a great question. You are correct regarding the accumulated mutations; there might be multiple changes at a site along an edge. We didn't highlight this in our paper, but the JC69 model considers such cases. The $b=f(p)$ formula in Eqn.12 is the corrected distance formula of the JC69 model; hence our JC Sampler considers accumulated mutations by default. The independence assumption is common in all nucleotide substitution models, and dependence cannot be considered in VaiPhy. With that said, a possible extension to the single-cell domain would be an exciting project (see also the answer above).
>
> ### Questions - Slantis
> > The separation between previous work and proposed approach is unclear for SLANTIS. The authors could perhaps make a better distinction between the two in the document. Due to the probabilistic nature of SLANTIS, is it possible that the leaves are not always all included in the reconstructed tree? If yes, how does this affect the performance?
>
> Thank you for the suggestion, we will add a statement to line 149 regarding SLANTIS' connection to [4]. In [4], the candidate edges are selected from those that will make a perfect matching and the probability of each edge is uniform. On the contrary, we make sure to have the spanning trees, and the probabilities of the edges are proportional to the weights of the trees.
> Regarding the leaves not appearing in the reconstructed tree: no, the leaves are always present in the sampled tree, as the final step of the SLANTIS algorithm is to sample leaf connections, see lines 15-22 in Alg.3 in Appendix B.
>
> [4] Diaconis, Persi. “Sequential Importance Sampling for Estimating the Number of Perfect Matchings in Bipartite Graphs: An Ongoing Conversation with Laci.”

---

> ### Author Response · Authors · 2022-08-02
> **Response to Reviewer Mtqk - Part 2/2**
>
> ### Questions - Parameterized CSMC
> > I believe that using VaiPhy to parameterize CSMC instead of using directly VaiPhy is impactful ... the number of taxa is larger than 100?
>
> Yes, the memory and run-time advantages of VaiPhy should be beneficial for the larger datasets. In the paper, we preferred to focus on the datasets used by the previous VI publications, but we are currently investigating larger datasets.
>
> ### Questions - Experiments
> > I think that the experiments are interesting but I would like to better understand how the proposed approach performs and scales in terms of number of taxa or sites for example compared to the baselines.
> Figure 1, which compares the number of mutations with branch length, is an interesting plot. I was wondering if it would be possible to also evaluate the reconstructed tree. This could be done on simulated data where the tree (and sequences of the ancestors) are known but only the leaves are used for phylogeny reconstruction.
>
> As in the VBPI-NF paper, we run MrBayes for a fixed number of iterations for each dataset, i.e., regardless of the number of taxa. This however, benefits MrBayes in two ways:
> 1. The upward bias in the Harmonic Mean estimator of the marginal likelihood is well-known and accentuated by a small support of posterior estimate obtained by the MCMC. Although this effect probably is reduced by SS, the same mechanism may be at play, which is an interesting research question in its own. The larger tree space associated with an increased number of taxa will be relatively less explored with the same number of iterations. Consequently, it is unclear how to perform a fair simultaneous comparison of time and accuracy.
> 2. The runtime scales linearly with the number of taxa.
>
> However, we examined the effect of different sequence lengths. We performed additional runtime experiments to investigate how VaiPhy and other benchmarks scale. In the tables below, we put the runtimes for two datasets; a subset of the DS1 (27 taxa, 1000 sites) and the DS1 (27 taxa, 1949 sites). The runtimes are averaged over several repetitions. For the 8 and 16 core runs, we increased the number of chains per MrBayes SS run (a detailed motivation is provided in our answer to the Reviewer vFFt's limitations section). MrBayes SS is the fastest method when multiple cores are enabled. MrBayes SS' runtime increased linearly with the number of sites.
> We were unable to run VCSMC (for some number of cores) due to memory issues. It is interesting to note that VBPI-NF's architecture, hence its runtime depends on the number of unique splits provided as input. Moreover, the number of VBPI-NF parameters was higher for the smaller sequence length (10 million vs 8.4 million). Independently, VBPI-NF was clearly slower than the other methods.
>
> **Table 2** Wallclock runtimes in seconds for a _modified_ DS1 dataset of size (27, 1000). MrBayes SS has the following configurations; (number of chains, number of cores) $\{(4,1), (4,2), (4,4), (8,8), (16,16)\}$.
> | Cores | MrBayes SS | VBPI-NF | VaiPhy | VCSMC |
> | ---: | ---: |  ---: | ---: | ---: |
> |  1 | 2703 | still running (> 30 hours / 108000 sec) | 4487 | out of memory |
> |  2 | 1690 | still running (> 30 hours / 108000 sec) | 3906 | out of memory |
> |  4 | 1244 | still running (> 30 hours / 108000 sec) | 3828 | out of memory |
> |  8 | 1330 | still running (> 30 hours / 108000 sec) | 3717 | out of memory |
> | 16 | 1146 | still running (> 30 hours / 108000 sec) | 3629 | 3248 |
>
>
> **Table 3** Wallclock runtimes in seconds for the DS1 dataset of size (27, 1949). MrBayes SS has the following configurations; (number of chains, number of cores) $\{(4,1), (4,2), (4,4), (8,8), (16,16)\}$.
> | Cores | MrBayes SS | VBPI-NF | VaiPhy | VCSMC |
> | ---: | ---: |  ---: | ---: | ---: |
> |  1 | 5859 | 128522 | 4841 | out of memory |
> |  2 | 3487 | 128586 | 4263 | out of memory |
> |  4 | 2137 | 114953 | 3744 | out of memory |
> |  8 | 2200 | 109651 | 3969 | out of memory |
> | 16 | 2147 | 106914 | 3848 | 6840 |
>
> In addition, following your suggestion, we tried to illuminate the performance of SLANTIS in a fashion similar to Fig.1 in the paper. For some synthetic datasets, given the true topology, we computed the ancestral sequences on the internal vertices. We used those sequences to compute the pairwise edge weight matrix, $W$, and sampled $100$ tree topologies using SLANTIS. We plotted the histograms of the Robinson-Foulds distances between the samples and the true topology for several datasets. As shown in the plots, SLANTIS typically samples a large number of trees that have small RF distances to the correct topology. You can access the figure via this link: [https://imgur.com/a/W5HlIWr](https://imgur.com/a/W5HlIWr).
>
> ### Minor comments
> Thank you for providing a list of items to improve our paper. We modified the manuscript accordingly; the main content change was the updated version of lines 153-154. In Eqn.7, $\tau$ appears in the summation term.

---

> > ### Comment · Reviewer_Mtqk · 2022-08-08
> > **Answer to authors**
> >
> > I thank the authors for their thorough response. The authors addressed my questions and comments. In light of the authors's answers to my review and to the other reviews, I am in favor of accepting the paper at NeurIPS and increase the grade to 6.

---

### Official Review · Reviewer_k9KB · 2022-07-10

**Rating:** 7
**Confidence:** 3
**Soundness:** 2 fair
**Presentation:** 4 excellent
**Contribution:** 3 good

**Summary:**

Inferring the posteriors of phylogenetic trees is critical to evaluating uncertainty in tree inference. The most popular method for this is MCMC; however, in some situations effective sampling of the combinatorial tree space can be difficult, and MCMC does not return (straightforwardly) densities for individual trees, meaning that it can have difficulties representing diffuse posteriors. Variational inference, in principle, can solve these issues while decreasing the computational cost of inference. However current methods require lots of resources for training due to the need to perform automatic differentiation.

In this paper, the authors aim to improve the efficiency of variational inference. They make a number of contributions to do so:
1) They introduce an efficient algorithm to sample trees conditional on branch lengths, substitution probabilities, and sequences at every vertex.
2) For a Jukes-Cantor substitution model, they introduce an algorithm to sample branch lengths given the topology of the tree and sequences at every vertex.
3) They introduce an algorithm for iteratively updating the posterior of the sequences at each vertex of the tree, the tree topology, and the branch lengths. They calculate the natural gradients for this algorithm to make training as efficient as possible.
4) Finally, they also combine 1 and 2 to create a more efficient proposal for a tree sampling procedure based on stochastic monte carlo (SMC).

They demonstrate the efficiency of their method, taking substantially less time than another variational inference method and some sampling methods. They also demonstrate that their method in 4 indeed increases the efficiency of sampling.

**Questions:**

1) For the transformed branch length, of course when there are many sites and branch lengths are long this won't matter, what do you do when p<1/4? Do you account for p's that cannot give rise to valid branch lengths when calculating probabilities?
2) Why did you compare to VCSMC instead of VNCSMC, the improved model? And did you perform variational inference on the parameters in phi-CSMC?
3) Did you use the same branch length priors for all models (I believe other models have exponential branch length priors by default)?
4) The likelihood estimation methods used for the VI and SMC methods are biased downwards and so not directly comparable. This lower bound is still useful to report for the SMC methods as it demonstrates more efficient sampling when it is higher, but this isn't a fair comparison to VSMC I believe which also optimises the parameters of its branch length priors and substitution matrix. However this isn't a fair comparison to MCMC which doesn't have a downward-biased marginal likelihood estimate. In other phylogenetics VI papers, samples are usually taken from the VI methods and their marginal likelihood calculated unbiasedly just as with the samples from MCMC. This could also be done for your method I believe, allowing evaluation without the inference of the sequences at internal nodes affecting the bias.


**Limitations:**

1) The mean field approximation to internal nodes may severely limit the ability of this model to fit certain posteriors.
2) This method requires a very efficient way to sample branch lengths, which the authors demonstrate exists for the JC model, but may be more difficult to devise for more complicated substitution models.

**Strengths And Weaknesses:**

Strengths:
1) The paper proposes an extremely efficient method for variational inference of phylogenetic trees.
2) In the process of devising this method, the authors propose a new method to sample trees topologies and brnch lengths that is of independent interest.
These points make it a substantial contribution to phylogenetic inference

Weaknesses:
1) It is obvious that classical variational inference methods are extremely inefficient (when first reported these methods were shown to be more efficient that MCMC based on "number of iterations needed to reach a minimum accuracy", however, as iterations come with different computational costs, the method the authors used to measure efficiency here is more sound.) However the number of iterations to run the classical method in this case seem to be arbitrary, or selected to be very large in the original publication to get to convergence. It would be more fair to compare compute time to log likelihood achieved.
This concern is more dire in the comparison between CSMC methods, the author's proposed method and MCMC - although not strictly necessary to demonstrate a contribution as one obtains different things from each methods.
2) I am not 100% on the likelihood evaluations (see questions below).

---

> ### Author Response · Authors · 2022-08-02
> **Response to Reviewer k9KB**
>
>
> ### Weaknesses - Comparison
> > It is obvious that classical variational inference methods are extremely inefficient (when first reported these methods were shown to be more efficient that MCMC based on "number of iterations needed to reach a minimum accuracy", ... This concern is more dire in the comparison between CSMC methods, the author's proposed method and MCMC - although not strictly necessary to demonstrate a contribution as one obtains different things from each methods.
>
> We performed a similar analysis with VBPI-NF (see lines 280-281). We agree that the comparisons with the CSMC and MCMC methods are harder to perform. It is less critical due to the performance of the VCSMC method, that is we achieve a higher lower bound throughout the optimization.
>
> ### Questions - Transformed branch lengths
> > For the transformed branch length, of course when there are many sites and branch lengths are long this won't matter, what do you do when $p<1/4$? Do you account for p's that cannot give rise to valid branch lengths when calculating probabilities?
>
> When $p=0.25$, the sequences are _saturated_. A pair of random sequences, that is, independent sequences with uniformly chosen nucleotides, have an expected $p$ of $0.25$. Under the JC model, when the evolutionary time between a pair of sequences approaches infinity, they become a pair of random sequences. In conclusion, $p<0.25$ is not an interesting case.
> Also as you noted, this is not a big problem in the context of the long sequences, as the expected number of non-mutations between two nodes will be significantly larger than the opposite. One of the features of the JC sampler, which is unexplored in this paper, is the usefulness of the Beta prior parameters. In extremely low-data regimes, $p>0.25$ may not be ensured when using a uniform prior. Then, we can easily use the prior parameters to skew the posterior distribution away from $p<0.25$. A thorough analysis of this is the subject  of our ongoing future work.
> In practice, on the rare occasions when we encounter $p<0.25$ within our JC sampler, we simply discard that sample and draw another $p$ from the Beta posterior distribution. We added a section about this (Appendix C).
>
> ### Questions - VNCSMC
> > Why did you compare to VCSMC instead of VNCSMC, the improved model?
>
> Thank you for the question. We used the authors' GitHub repository specified in their paper, but the VNCSMC code was not functional. It gave rise to non-trivial memory issues that couldn't be resolved with reasonable changes to the code. We would also like to direct you to our updated discussion in Appendix H regarding the reproducibility of the results.
>
> ### Questions - $\phi$-CSMC
> > And did you perform variational inference on the parameters in phi-CSMC?
>
> The proposals used in $\phi$-CSMC are learned from VaiPhy. It is learned once in VaiPhy before the execution of $\phi$-CSMC, and used only to parameterize the proposals in the $\phi$-CSMC.
>
> ### Questions - Branch Length Priors
> > Did you use the same branch length priors for all models (I believe other models have exponential branch length priors y fault)?
>
> Yes, for a fair comparison, we used the same branch length prior, $Exp(10)$, for all variational methods. We picked the same rate parameter as in the VBPI-NF paper.
>
> ### Questions - Bias
> > The likelihood estimation methods used for the VI and SMC methods are biased downwards and so not directly comparable. This lower bound is still useful ... This could also be done for your method I believe, allowing evaluation without the inference of the sequences at internal nodes affecting the bias.
>
> When evaluating the IWELBO, we do not use our variational distribution over ancestral sequences, as can be seen in Eqn.15 in Section 3.4. However, as you pointed out, we are comparing estimators biased downwards with estimators that potentially may have another bias. As you also pointed out, this is to our disadvantage. This clearly motivates further research, for instance, what would be the impact of applying the SS method in the VI context.
>
> ### Limitations - Mean-Field Approximation
> > The mean field approximation to internal nodes may severely limit the ability of this model to fit certain posteriors.
>
> This is a good point. We are already working on a different model with other variational distributions for the same problem setting.
>
> ### Limitations - Branch Length Sampling for Other Substitution Models
> > This method requires a very efficient way to sample branch lengths, which the authors demonstrate exists for the JC model, but may be more difficult to devise for more complicated substitution models.
>
> Thank you for this great question. JC69 is one of the most commonly used substitution models; therefore, it was our focus in this project. We have; however, successfully expanded the methodology to two more complex substitution models; we are actively working on a manuscript that describes these samplers and includes additional analysis.

---

> > ### Comment · Reviewer_k9KB · 2022-08-03
> > **Response to author comments**
> >
> > I thank the authors for their thorough response.
> >
> > The authors address the limitations I mentioned.
> >
> > The authors point out, with respect to weakness 1, that performing the analysis I mentioned is unlikely to change their results. I agree with their reasoning. The result is similar for weakness 2, where a change in the analysis would be insightful, but would not change their results.
> >
> > The authors addressed all my questions as well.

---

### Official Review · Reviewer_vFFt · 2022-07-22

**Rating:** 8
**Confidence:** 4
**Soundness:** 4 excellent
**Presentation:** 3 good
**Contribution:** 4 excellent

**Summary:**

This paper contributes to the existing Bayesian phylogenetic VI literature with three novel methods which, in combination, lead to better accuracy, lower memory requirements, and faster computation speed when compared to existing VI methods in the space. We now discuss each of the authors' three novel contributions in isolation.

SLANTIS, a method for generating proposal trees in the multifurcating tree space, is the first contribution. As reported, this method is orders of magnitude faster than existing methods for generating proposal trees, taking far fewer iterations to approximate the relevant expectations than existing methods.

JC sampler uses the computed matrix of expected mutations across branches to sample branch lengths exactly according to the Jukes Cantor model. Sampling branch lengths directly from the underlying substitution model instead of from an arbitrary prior just makes sense, and frankly I'm surprised that this is the first paper to do so.

VaiPhy is the full variational inference method developed in this paper, utilizing both SLANTIS and JC sampler. VaiPhy enjoys significantly lower computational times than existing VI methods, while consistently producing better log-likelihood scores. Additionally, VaiPhy includes an auxiliary variable for ancestral sequences, which helps pave the way for potential future applications which may want to include ancestral data, such as that of fossils.

Bayesian phylogenetic VI is a new and growing methodology, so, while the methods in this paper do not produce the best results among all phylogenetic inference methods, they provide significant contributions towards the development of VI in phylogenetics.

**Questions:**

Section 3.4 states that "the framework learns its parameters and distributions by maximizing the ELBO [... but] the IWELBO offers a tighter lower bound" and "using SLANTIS and the JC sampler we can easily evaluate our framework using the IWELBO." This statement is confusing: the former states that the ELBO is used in the framework, but the latter seems to imply that the IWELBO is used instead; which is true?

Have you considered utilizing ancestral sequence data (e.g. from fossils) as inputs into the method, instead of featuring them exclusively as auxillary variables?

Why were convergence statistics for SLANTIS not included? When proposing a method which proports to require orders of magnitude fewer iterations than existing methods, convergence seems an important attribute to investigate.

The method improves on memory requirementns of previous VI implementations by utilizing significantly fewer learnable parameters, but what amount of memory does the method actually use at runtime? I.e. how many GiB of ram does the method utilize? This question is very relevant to the current limitations of the method.

**Limitations:**

Of all phylogenetic tree inference methods, this is neither the fastest nor the most accurate. The authors very honestly admit this, and though this is a limitation, it is should _not_ be considered a weakness of the paper, as  VI in Bayesian inference is still in its infancy, while their competitor MrBayes is comparatively a full grown adult. That being said, MrBayes should not have been excluded from the wallclock time vs. number of cores plots, as how far modern VI methods lag behind MrBayes is not irrelevant.

A very typical assumption to make in phylogenetic likelihood computations is independence of branch lengths. Computations without this assumption are rarely feasible. This method calculates branch lengths in isolation, though, without respect to topology, which I expect is far too much of a simplifying assumption.

Log-likelihood values for the method (as well as its competitors) are provided on 7 real-world datasets, but this metric is not actually very good for evaluating the performance of phylogenetic inference. Better log-likelihood scores do not _necessarily_ equate to a better estimate of the underlying tree, especially when 1) many simplifying assumptions are made in likelihood calculations, and 2) underlying models (such as the substitution model, JC in this case) may very well be misspecified. A better way to evaluate the performance of these methods is through a simulation study where the ground truth trees are known, and prediction accuracy can then be calculated via the Robinson-Foulds distance.

**Strengths And Weaknesses:**

## Originality

While the classical VI methodology that underlies this paper's work is nothing new, SLANTIS and JC sampler are both completely novel methods. Additonally, ancestral sequence data has been included in previous Bayesian methods as part of the input, not just as auxillary variables. However, VaiPhy is the first VI-based method to take ancestral data into consideration, even if only probabilistically. Thus, this paper contributes a significant body of original material towards the improvement of VI-based phylogenetic inference.

## Quality

The overall quality of the methods presented is very high. Further details are contained in other relevant sections.

## Clarity

The methodology is explained very well; as someone familiar both with phylogenetics and variational inference I felt that I had a very good understanding of the mechanisms behind VaiPhy on my first read, which is not an easy task for a paper on something as computationally heavy as VI.

The background section of the paper is devoid of any explanation of sequential Monte Carlo or conditional sequential Monte Carlo, which is a barrier for readers who are interested in the potential phylogenetic applications of the work, but unfamiliar with these concepts. Additionally, while SMC is at least given an introduction, CSMC is not (in fact the word "conditional" does not appear in the paper). While the paper is no doubt intended to appeal to an audience which is already familiar with variational inference, its applications are in phylogenetics, so these concepts should still be given an adequate introduction and brief explanation.

## Significance

VI is a developing field in phylogenetic inference. I do not know whether VI will end up leading the cutting edge in phylogenetics, but, irregardless, this paper's contributions certainly make that reality seem more possible.

---

> ### Author Response · Authors · 2022-08-02
> **Response to Reviewer vFFt - Part 1/2**
>
>
> ### Clarity - SMC and CSMC
> > The background section of the paper is devoid of any explanation of sequential Monte Carlo or conditional sequential Monte Carlo, which is a barrier for readers who are interested in the potential phylogenetic applications of the work, but unfamiliar with these concepts. Additionally, while SMC is at least given an introduction, CSMC is not (in fact the word "conditional" does not appear in the paper). While the paper is no doubt intended to appeal to an audience which is already familiar with variational inference, its applications are in phylogenetics, so these concepts should still be given an adequate introduction and brief explanation.
>
> Thank you for pointing this out. We will add more details regarding SMC and CSMC to the camera-ready version of the paper.
>
> ### Questions - ELBO and IWELBO
> > Section 3.4 states that "the framework learns its parameters and distributions by maximizing the ELBO [... but] the IWELBO offers a tighter lower bound" and "using SLANTIS and the JC sampler we can easily evaluate our framework using the IWELBO." This statement is confusing: the former states that the ELBO is used in the framework, but the latter seems to imply that the IWELBO is used instead; which is true?
>
> The ELBO is maximized in order to learn the model parameters ($\phi$ and the parameters of the distribution $q(Z|X)$. Meanwhile, the IWELBO, an estimator of the marginal log-likelihood, is used as an evaluation metric (i.e., not a training objective). This is a standard approach in modern VI (for instance, see [1] and [2]).
>
> [1] Tomczak, Jakub, and Max Welling. 2018. “VAE with a VampPrior.”
>
> [2] Vahdat, and Kautz. n.d. “NVAE: A Deep Hierarchical Variational Autoencoder.”
>
> ### Questions - Utilizing Ancestral Sequences
> > Have you considered utilizing ancestral sequence data (e.g. from fossils) as inputs into the method, instead of featuring them exclusively as auxillary variables?
>
> This is an interesting suggestion. We didn't investigate the use of ancestral sequences as input; however, our framework can handle such data with minor adjustments. We can compute the edge weights using the _observed_ ancestral sequences (Eqn.9), and SLANTIS can sample trees with observed ancestral nodes.
>
> ### Questions - Convergence of SLANTIS
> > Why were convergence statistics for SLANTIS not included? When proposing a method which proports to require orders of magnitude fewer iterations than existing methods, convergence seems an important attribute to investigate.
>
> This is an interesting question. However, the support of the induced SLANTIS distribution contains all of the exponentially many spanning trees of the complete graph associated with the weight matrix, so we can only hope for convergence to a useful approximation of this distribution. We have made a small investigation that highlights that SLANTIS finds a diverse set of trees in the neighborhood of the true tree with respect to the Robinson-Foulds metric (see the answer to Reviewer Mtqk's question).
>
>
> ### Questions - Memory
> > The method improves on memory requirementns of previous VI implementations by utilizing significantly fewer learnable parameters, but what amount of memory does the method actually use at runtime? I.e. how many GiB of ram does the method utilize? This question is very relevant to the current limitations of the method.
>
> Thank you for identifying a relevant but missing perspective in our analysis. We conducted an experiment where we measured the memory used by the Python process during the runtime of VaiPhy and compared it to VCSMC on datasets DS1 mini (27 taxa, 1000 sites) and DS1 (27 taxa, 1949 sites). The results are presented in the table below. Notably, VaiPhy is in the order of MB and doesn't scale with number of sites while VCSMC requires many GBs.
>
> **Table 1** Runtime memory usage (MB).
> | Dataset (taxa, sites) | VaiPhy | VCSMC |
> | ---: | ---: |  ---: |
> | DS1 mini (27, 1000) | 286 | 21935 |
> | DS1 (27, 1949)      | 285 | 23060 |

---

> ### Author Response · Authors · 2022-08-02
> **Response to Reviewer vFFt - Part 2/2**
>
> ### Limitations - MrBayes Runtime
> > Of all phylogenetic tree inference methods, this is neither the fastest nor the most accurate. The authors very honestly admit this, and though this is a limitation, it is should not be considered a weakness of the paper, as VI in Bayesian inference is still in its infancy, while their competitor MrBayes is comparatively a full grown adult. That being said, MrBayes should not have been excluded from the wallclock time vs. number of cores plots, as how far modern VI methods lag behind MrBayes is not irrelevant.
>
> We followed the experimental setup in [3] for MrBayes SS. Each MrBayes run contains 4 chains. The parallel MrBayes software requires the number of cores to be greater than or equal to the number of chains. Therefore, we didn't include the MrBayes SS results in Fig.3 (since we need to change the number of chains per run for the different number of cores).
> However, to provide you an insight, we performed some additional experiments for the following MrBayes SS settings (number of chains, number of cores) $\{(4,1), (4,2), (4,4), (8,8), (16,16)\}$ on the DS1 dataset and obtained the following average runtimes in seconds over five repetitions respectively; $\{5859, 3487,  2137, 2200, 2147\}$. We can add a statement in the runtime results section and add the complete results of 2 and 4 cores run to Fig.3; however, we prefer not to include 8 and 16 cores results.
> VaiPhy runs on a single core; however, there are some parts that trivially can be parallelized within our algorithm (e.g., sampling with SLANTIS and JC Sampler) and so that it becomes at least as fast MrBayes with multiple cores.
>
> [3] Zhang, Cheng. 2020. “Improved Variational Bayesian Phylogenetic Inference with Normalizing Flows.”
>
> ### Limitations - Independence of Branch Lengths
> > A very typical assumption to make in phylogenetic likelihood computations is independence of branch lengths. Computations without this assumption are rarely feasible. This method calculates branch lengths in isolation, though, without respect to topology, which I expect is far too much of a simplifying assumption.
>
> We agree that a dependent way of sampling the branch lengths should improve our results. We will add a statement to the conclusion about the future directions. In fact, we are following this line of research for the purpose of the follow-up paper.
>
> ### Limitations - Robinson-Foulds Distance
> > Log-likelihood values for the method (as well as its competitors) are provided on 7 real-world datasets, but this metric is not actually very good for evaluating the performance of phylogenetic inference. Better log-likelihood scores do not necessarily equate to a better estimate of the underlying tree, especially when 1) many simplifying assumptions are made in likelihood calculations, and 2) underlying models (such as the substitution model, JC in this case) may very well be misspecified. A better way to evaluate the performance of these methods is through a simulation study where the ground truth trees are known, and prediction accuracy can then be calculated via the Robinson-Foulds distance.
>
> Thank you for the suggestion. The marginal loglikelihood comparisons are one of the key results for the methods we compare. We wanted to show how VaiPhy performs using a frequently used metric, and investigate the reproducibility of existing methods. The RF distance hasn't been used in state-of-the-art VI papers, probably because RF is a maximum likelihood-oriented metric; we aim for model selection and comparison with other VI methods using evaluation measures they have targeted.

---

### Author Response · Authors · 2022-08-02
**General Response**

We thank the reviewers for their insightful comments, which have helped us to improve our work. We have addressed your comments and concerns with clarifications and additional experiments and updated the manuscript accordingly. Due to the page and time limit, we were unable to include some of the suggestions to the revised manuscript; however, we will use the extra page in the camera-ready version to incorporate them in the case of acceptance.

---

### Meta-Review · Area_Chair_8sBp · 2022-08-27

**Recommendation:** Accept
**Confidence:** Certain

**Metareview:**

The authors propose a novel variational inference method for phylogenetic inference with a specific focus towards improving the speed of variational inference. The reviewers agree that the paper makes significant contributions to an important problem. The authors are encouraged to take the reviewer feedback carefully in preparing the next version of the manuscript.

**Award:**

No

---

### Decision · Program_Chairs · 2022-09-14

Accept